# Visceral Arterial Pseudoaneurysms—A Clinical Review

**DOI:** 10.3390/medicina61071312

**Published:** 2025-07-21

**Authors:** Ashita Ashish Sule, Shreya Sah, Justin Kwan, Sundeep Punamiya, Vishal G. Shelat

**Affiliations:** 1Yong Loo Lin School of Medicine, National University of Singapore, 10 Medical Drive, Singapore 117597, Singapore; ashita.ashish02@gmail.com; 2Department of Medicine, University of New South Wales Sydney, Kensington, Sydney, NSW 2033, Australia; 3Department of Vascular and Interventional Radiology, Tan Tock Seng Hospital, 11 Jalan Tan Tock Seng, Singapore 308433, Singapore; justinkwan07@gmail.com (J.K.); punamiya@gmail.com (S.P.); 4Department of Hepato-Pancreatico-Biliary Surgery, Tan Tock Seng Hospital, 11 Jalan Tan Tock Seng, Singapore 308433, Singapore; vgshelat@gmail.com

**Keywords:** doppler, embolization, endovascular, pseudoaneurysm, rupture, surgery

## Abstract

*Background and Objectives*: Visceral arterial pseudoaneurysms (VAPAs) are rare vascular lesions characterized by the disruption of partial disruption of the arterial wall, most commonly involving the intima and media. They have an estimated incidence of 0.1–0.2%, with the splenic artery most commonly affected. Their management poses unique challenges due to the high risk of rupture. Timely recognition is crucial, as unmanaged pseudoaneurysms have a mortality rate of 90%. This narrative review aims to synthesize current knowledge regarding the epidemiology, etiology, clinical presentation, diagnostic methods, and management strategies for VAPAs. *Materials and Methods*: A literature search was performed across Pubmed for articles reporting on VAPAs, including case reports, review articles, and cohort studies, with inclusion of manuscripts that were up to (date). VAPAs are grouped by embryological origin—foregut, midgut, and hindgut. *Results*: Chronic pancreatitis is a primary cause of VAPAs, with the splenic artery being involved in 60–65% of cases. Other causes include acute pancreatitis, as well as iatrogenic trauma from surgeries, trauma, infections, drug use, and vascular diseases. VAPAs often present as abdominal pain upon rupture, with symptoms like nausea, vomiting, and gastrointestinal hemorrhage. Unruptured pseudoaneurysms may manifest as pulsatile masses or bruits but are frequently asymptomatic and discovered incidentally. Diagnosis relies on both non-invasive imaging techniques, such as CT angiography and Doppler ultrasound, and invasive methods like digital subtraction angiography, which remains the gold standard for detailed evaluation and treatment. A range of management options exists that are tailored to individual cases based on the aneurysm’s characteristics and patient-specific factors. This encompasses both surgical and endovascular approaches, with a growing preference for minimally invasive techniques due to lower associated morbidity. *Conclusions*: VAPAs are a critical condition requiring prompt early recognition and intervention. This review highlights the need for ongoing research to improve diagnostic accuracy and refine treatment protocols, enhancing patient outcomes in this challenging domain of vascular surgery.

## 1. Introduction

### 1.1. Definition

Unlike true aneurysms which involve the thinning of all arterial wall layers, visceral arterial pseudoaneurysms (VAPAs) commonly involve the disruption of two layers (the intima and media) [1], resulting in an enclosed rupture and consequently a peri-artery hematoma [2].

### 1.2. Epidemiology

#### 1.2.1. Incidence

VAPAs are rare occurrences, with a reported incidence of 0.1–0.2%, making them rare entities. This is, however, likely to be an underestimation due to asymptomatic patients [1].

The most commonly affected artery is the splenic artery followed by the hepatic, superior mesenteric, celiac, gastric and gastroepiploic, intestinal (jejunal, ileal, and colic), pancreatic and pancreaticoduodenal, gastroduodenal, and inferior mesenteric arteries [3]. Although each visceral artery territory exhibits distinct clinical and anatomical characteristics, pseudoaneurysms across these territories share common themes in terms of pathogenesis and risk.

#### 1.2.2. Demographics

Certain VAPAs such as those of the right gastroepiploic and jejunal arteries have been found to more commonly affect men and older people [4,5]. Patient factors such as atherosclerosis, hypertension, bleeding disorders, and vasculitis could predispose to VAPA formation [6,7,8,9,10].

#### 1.2.3. Complications

VAPAs are associated with a higher risk of rupture, morbidity, mortality, and reintervention rates compared to true aneurysms [11,12,13]. Complications such as ischemia, coil migration, contrast nephropathy, non-target embolization, and stent thrombosis may also arise from the treatment. Furthermore, their embryological origin has practical implications in patient-centered outcomes. For instance, pseudoaneurysms involving midgut circulation may necessitate small bowel resections, which carry a risk of short bowel syndrome associated with significant nutritional and metabolic challenges. Similarly, hindgut pseudoaneurysms may result in segmental colectomy or low anterior resections, potentially requiring a stoma. Both outcomes have profound implications on patients’ quality of life and long-term psychosocial well-being, underscoring the importance of an anatomical framework in guiding both clinical decisions and patient counseling.

### 1.3. Etiology

Pancreatitis (mostly chronic) [5,14,15,16] is a common cause of VAPAs, particularly foregut VAPAs such as those of the splenic, gastroduodenal, and superior mesenteric arteries [16,17]. For pseudoaneurysms caused by chronic pancreatitis, the splenic artery is the most commonly involved (60–65%), followed by gastroduodenal (20–25%), pancreaticoduodenal (10–15%), hepatic (5–10%), and left gastric arteries (2–5%) [18]. Pseudoaneurysms due to acute pancreatitis arise from autodigestion and weakening of the walls of nearby vessels by proteolytic pancreatic enzymes [19]. Pseudoaneurysms caused by both acute and chronic pancreatitis are important to recognize as they are associated with a mortality of 90% if not recognized and managed [19].

Increasingly, iatrogenic causes have become more common with the heightened use of percutaneous endovascular interventions [20]. The majority of the pseudoaneurysms associated with surgery have been reported to occur after hepatobiliary and pancreatic surgeries, which are posited to arise as a result of intraoperative mechanical vascular injuries, pancreatic enzyme leaks, bile leaks, enteric fistulae, and intraabdominal abscesses with sepsis [21].

Other causes include non-iatrogenic trauma [5,22]; ethanol abuse [5,23]; infections such as infective endocarditis, cholecystitis, and atherosclerosis [22]; drugs such as apixaban [24]; collagen vascular diseases; peptic ulcer disease; vascular abnormalities such as fibromuscular dysplasia [22]; polyarteritis nodosa [5,25]; liver cirrhosis and congenital inflammatory diseases [22]; and spontaneous formation [23,26]. Figure 1 shows the estimated distribution of VAPA for various arteries. To provide greater clinical utility, we grouped the pseudoaneurysms anatomically and embryologically to reflect shared vascular behavior and interventional considerations. This understanding is crucial for anticipating rupture risk, selecting diagnostic modalities, and individualizing treatment plans.

### 1.4. Presentation

VAPAs commonly present as abdominal pain upon rupture [27]. Other symptoms include nausea, vomiting, hematemesis, melena, and hematochezia [28,29]. They are often asymptomatic and discovered incidentally [30]. Unruptured VAPAs can manifest as a pulsatile mass or bruit [31]. However, it more commonly presents as gastrointestinal hemorrhage upon rupture which, determined by its site, can result in hemobilia, intraperitoneal or retroperitoneal hemorrhage, and hemorrhagic shock [32,33]. Hemosuccus pancreaticus and gastric outlet obstruction are other possible clinical manifestations [28].

### 1.5. Diagnosis

#### 1.5.1. Overview

VAPAs can be diagnosed in both asymptomatic and symptomatic patients using non-invasive and invasive imaging methods. Non-invasive imaging techniques include computed tomography angiography (CTA), doppler ultrasound [34], magnetic resonance imaging (MRI), magnetic resonance angiography (MRA) [35], and multidetector computed tomography (MDCT) angiography scans [20,35]. Invasive imaging techniques include digital subtraction angiography (DSA) and conventional angiography [35,36].

Elevated levels of lactic acid or serum enzymes (e.g., amylase, lipase, etc.) may be present in the context of underlying conditions such as pancreatitis or hemorrhagic shock but are not specific indicators of pseudoaneurysm. These laboratory abnormalities should be interpreted alongside clinical and imaging findings.

#### 1.5.2. Computed Tomography (CT) and CTA

MDCT and CTA are high-resolution non-invasive modalities with a short breath-hold time that are useful for the visualization of contrast filling during the arterial phase [37,38]. They are the most frequently utilized and sensitive non-invasive methods for identifying pseudoaneurysms [35]. These methods also detail the location, size, severity, and surrounding effects of the aneurysm [39]. While diagnosis can be made using both CTA [40] and conventional angiograms, which are contrasted CT imaging and can be combined with therapeutic interventions, triple-phase CT angiography remains the investigation of choice, since it is non-invasive and can be performed without any preparation in the emergency setting [40,41,42]. It must be noted that a portal venous phase CT scan only can limit diagnosis, especially in patients with nonspecific symptoms, as an arterial phase CT scan is the most sensitive for detecting pseudoaneurysms and hemorrhage [43]. Additionally, pseudoaneurysms with a narrow neck might not be visible during the arterial phase and may only become apparent during the venous phase [35]. Metallic implants such as surgical drains or coils can cause artefacts and disguise the pseudoaneurysm [43]. In pancreatitis, risk stratification using a modified CT severity index score can allow unexpected outcomes and investigations to be planned for [44]. However, the use of CT is contraindicated in renal failure, poor intravenous access, and allergic reactions to contrast agents [38].

The risk of contrast-induced nephropathy (CIN) is important to recognize, as it is associated with significant morbidity through acute kidney failure [45]. Identifying patients at an increased risk of CIN is therefore important to mitigate adverse outcomes. This can be done by identifying risk factors in patients and calculating a CIN score, with the highest risk being in patients with chronic kidney disease and those with glomerular filtration rates (GFR) < 30 mL/min at the time of contrast administration [45,46,47]. The decision to proceed with imaging requiring contrast should then be based on clinical judgement of the physician who should account for the risk of CIN.

#### 1.5.3. DSA

While contrast-enhanced CT is more effective in detecting aneurysms in the main visceral branches, DSA can be useful for those located in small and peripheral branches [48]. DSA is the gold standard diagnostic, management, and follow-up modality, as it provides accurate details of the flow, site, and size of the pseudoaneurysm and allows for real-time assessment of contrast extravasation [28,38]. DSA’s high spatial resolution allows for imaging of small vessels, precise indication of size and anatomical location of aneurysms, and concurrent treatment. However, it can have complications associated with arterial puncture [38] and is only indicated for guiding embolization and when other imaging is normal despite high clinical suspicion [35,49].

#### 1.5.4. Doppler Ultrasound

The Doppler ultrasound modality can also help determine patency and distal flow [26,50]. It is the preferred screening modality for small pseudoaneurysms in which they commonly appear as a hypoechoic cystic structure, potentially with a characteristic “ying-yang sign” arising from the to-and-fro blood flow from the neck to the aneurysmal sac [34]. Doppler ultrasounds enable rapid, bedside imaging, particularly for patients in an unstable condition. It is radiation-free and hence useful in pregnancy, as well as is able to delineate the neck of the pseudoaneurysm [51,52]. However, it is operator-dependent, and its limited spatial resolution may cause small lesions to be missed. Its sensitivity is also limited by obesity, shadowing from bowel gas, and arteriosclerosis, and it is not sensitive for detecting active bleeding [52,53].

#### 1.5.5. MRA

MRA provides high spatial resolution but has lower sensitivity than CT angiography and is relatively contraindicated in patients with pacemakers or aneurysm clips, claustrophobia, or inability to hold their breath [54]. While MRA avoids exposure to radiation, it has limited utility in emergencies where patients are hemodynamically unstable [35,55].

#### 1.5.6. Newer Modalities

Emerging imaging technologies such as dual-energy CT (DECT) and contrast-enhanced ultrasound (CEUS) are increasingly being explored for their potential to enhance sensitivity in detecting small or complex pseudoaneurysms. These techniques may provide better visualization in anatomically challenging areas and could become useful adjuncts in early detection. The CEUS offers dynamic real-time imaging, making it especially useful in patients with contraindications to iodinated contrast. It is sensitive in detecting low-flow vascular lesions and has shown utility in surveillance following embolization. Its uptake remains limited by operator expertise and access to contrast microbubbles.

### 1.6. Management

#### 1.6.1. Overview

Unlike true aneurysms, every pseudoaneurysm should be treated as soon as possible, regardless of its size, as its risk of rupture is innate [16]. Management of VAPAs involves operative and non-operative management. Operative management includes open or laparoscopic procedures to clip, ligate, or excise the pseudoaneurysm. Non-operative management includes endovascular techniques such as angioembolization of the parent artery and thrombin injections [2,25,56,57].

While the conventional open surgical approach remains the gold standard for VAPAs, minimally invasive approaches are first line in stable patients. However, in the presence of hemodynamic instability, open surgery remains the preferred and definitive treatment, as endovascular procedures may not offer adequate hemostasis or rapid control in emergency situations [35,57,58]. Minimally invasive options for treatment of VAPAs include an endovascular, percutaneous, or endoscopic approach. Agents used for embolization include coils and stent grafts, which reduce blood flow; coils and liquid embolics (glue and thrombin), which stimulate thrombosis; and coils and liquid agents, which induce inflammation. Various factors need to be accounted for when deciding to embolize, namely, the size of the aneurysm and its neck, the territorial arterial oxygen supply, end-organ structure and function, the parent artery, the tortuosity and location of the artery, and the patient’s coagulation parameters [35]. Once an artery is occluded to treat the VAPA, the patient is at risk of ischemic complications, and this vulnerability must be communicated for patients to make a free and informed choice for the procedure.

#### 1.6.2. Endovascular Approach

An endovascular approach is the most prevalent and preferred means of embolization, with percutaneous or endoscopic techniques commonly considered only when the endovascular approach fails [14,59]. Endovascular treatment has shown low morbidity and mortality, as well as fewer complications, shorter hospital admissions, and quicker recovery in comparison to open surgery [25,60]. Access and expertise, the use of intravenous contrast agents, and the requirement of fluoroscopy imaging are essential requirements for this approach. Acute kidney injury, radiation from imaging, and dissection of the arterial wall remain inherent complications of this approach. Further, technical success does not always translate to clinical success, and ischemic complications may ensue. Thus, close follow-up monitoring of patients is essential.

Coils or microcoils are the most common agents used [57]. Coil embolization is preferred where vessel patency does not need to be preserved and can be employed using multiple methods and in combination with other interventional modalities. For example, coiling can be done prior to stent grafts, via super-selective microcatheters, alongside stenting, or just by itself [57]. Coiling techniques include sack packing, the sandwich technique, and proximal delivery [14,59]. Sack packing is indicated for saccular pseudoaneurysms with a narrow neck [14,59]. The sandwich technique is indicated for VAPAs with a collateral supply with occlusion done distally, across, and proximal to the neck, such as in splenic, hepatic, and gastroduodenal artery pseudoaneurysms [6], with success rates of over 90% [36,61,62]. End arteries such as renal arteries permit the proximal occlusion method [35,61]. The use of glue is uncommon due to greater associated complications such as non-target embolization, but it is useful in the case of high vessel tortuosity, pseudoaneurysm recurrence after the use of coils, and abnormal coagulation studies [30,36,63].

Stent graft (covered stent) placement, stent-assisted coiling, and balloon remodeling techniques reduce the migration of embolic material, particularly for wide-necked pseudoaneurysms, and promote maintenance of parent vessel patency and end-organ perfusion [14,31,57]. As navigating small, tortuous arteries is difficult with stent deployment, stent grafts are intended for larger proximal arterial segments, such as the common hepatic, splenic, and superior mesenteric arteries [30,64,65]. However, the increased flexibility and reduced size of novel stent grafts such as flow-diverting stents and self-expandable stents might mitigate this problem. Additionally, in VAPAs located at complex anatomical locations such as near vessel bifurcations, stents can be employed in a Y configuration [58,66,67]. In the case of an expandable parent artery, stent- or balloon-catheter-assisted coiling is used, where coil embolization can be guided through the gaps of a stent or by the side of a balloon, preventing the protrusion of coils into the vessel lumen [14,31]. However, their use is limited by great vessel tortuosity, inadequate vessel length on both sides of the aneurysm, as well as the risk of stent thrombosis and occlusion [57]. During a pancreaticoduodenectomy procedure, this consideration is integral in surgical conduct, and routinely, surgeons prefer to keep a slightly long stump of the gastroduodenal artery so that coil embolization is easier should a patient develop a pseudoaneurysm secondary to postoperative pancreatic fistula.

Other endovascular techniques include Gelfoam injection, vascular plug deployment, particle injection, and polyvinyl alcohol injection, which can be used in combination with coiling [57]. A vascular plug is an occlusive device that stimulates thrombosis with high accuracy and success rates, which is more cost-effective in instances where a lot of coils are needed [66,68].

Polyvinyl alcohol injections offer several benefits compared to glue such as their viscous consistency, slower polymerization rate, greater visibility under fluoroscopy, ability to evenly fill the vessel, and reduced tendency to stick to the microcatheter tip, which lowers the risk of non-target embolization. However, they are more expensive, require DMSO-compatible microcatheters, and take longer to prepare [66].

#### 1.6.3. Percutaneous Approach

A percutaneous approach, commonly done with ultrasonography, fluoroscopy, or CT guidance [66,69], is indicated for pseudoaneurysms enclosed by a solid organ or that are large with neighboring supporting structures and with vessels that are highly tortuous or sharply angulated, making endovascular access challenging [35,57]. It is associated with reduced morbidity and mortality compared to open surgery [70]. Embolic agents include thrombin, glue, and coils [35,57]. A thrombin injection can also be combined with endovascular coiling in a two-step approach [57]. Pseudoaneurysm rupture, non-target embolization, and recurrence are possible complications [35].

#### 1.6.4. Endoscopic Ultrasonography

Endoscopic ultrasonography (EUS) is indicated in specific cases such as splenic and gastroduodenal pseudoaneurysms, which can be detected via EUS when the endovascular approach has failed [71,72]. It involves the injection of thrombin or glue directly into the pseudoaneurysm, as guided by EUS [72].

#### 1.6.5. Open Surgery

Open surgery enables the end-organ condition to be ascertained visually and the need for revascularization to be assessed [30]. Open surgical repair of VAPAs includes excision, bypass, ligation, and end-organ resection [58]. This approach is indicated when other approaches are limited by patient anatomy (which is uncommon) and when the patency of the parent vessel is important and may be compromised with minimally invasive techniques [57]. Proximal and distal ligations are performed in emergency cases of rupture and in cases with sufficient collateral circulation, such as for splenic, celiac, and common hepatic arteries [30]. Figure 2 shows the flow chart of the diagnosis and management of suspected VAPA cases.

#### 1.6.6. Morbidity

Aside from perioperative morbidity, the following complications can happen following treatment, and a patient needs to be monitored and managed for ensuring good clinical outcomes:

##### Ischemia

Ischemic complications are among the most critical concerns following embolization of visceral pseudoaneurysms. Occlusion of the parent artery can compromise end-organ perfusion, particularly in vascular territories lacking robust collateral supply. This risk is heightened when embolization is performed in end arteries (e.g., renal, jejunal, or splenic branches). Clinical outcomes depend on accurate vessel selection and technique; hence, risk–benefit discussions must precede intervention, and postprocedural surveillance for ischemia is essential.

##### Coil Migration

Coil migration refers to the dislodgement of embolization coils from the target site into unintended vascular or enteric locations. This can lead to vessel thrombosis, end-organ embolization, or erosion into adjacent hollow viscera, potentially resulting in fistulas or delayed bleeding. Migration is more likely in wide-necked pseudoaneurysms or with inadequate packing. Techniques such as balloon remodeling or stent-assisted coiling can mitigate this risk.

##### Contrast-Induced Nephropathy (CIN)

The use of iodinated contrast agents in endovascular procedures carries a risk of CIN, particularly in patients with pre-existing renal insufficiency, diabetes, or volume depletion. CIN may present as an acute rise in serum creatinine within 48–72 h following a procedure. Preventive measures include pre-procedural hydration, the use of iso-osmolar contrast agents, and minimizing the total contrast volume. The CIN risk should be considered when choosing between CTA, DSA, and CEUS.

##### Non-Target Embolization

Non-target embolization occurs when embolic material inadvertently occludes adjacent vessels not supplying the pseudoaneurysm. This can cause unintended ischemia or infarction of surrounding tissues. The risk increases with liquid embolics like glue or when the catheter position is suboptimal. Super-selective catheterization and real-time imaging are essential to reduce this complication. The use of flow control techniques or plug-and-coil combinations can enhance precision.

##### Stent Thrombosis

Although stent grafts are valuable in maintaining vessel patency while excluding the pseudoaneurysm, they carry a risk of thrombosis, particularly in tortuous or small-diameter arteries. Stent thrombosis can lead to sudden ischemia of the dependent organs. Antiplatelet therapy is often required after the procedure, though this may conflict with concurrent bleeding risks. Regular imaging follow-up is essential to detect early occlusion or in-stent restenosis.

#### 1.6.7. Literature Search and Review Framework

This review was conducted using a narrative approach, with structured efforts to ensure transparency and reproducibility. Although not a systematic review, we incorporated adapted principles from the PRISMA framework to enhance methodological clarity. A literature search was performed in PubMed using combinations of the keywords “visceral pseudoaneurysm,” “splanchnic aneurysm,” “embolization,” “rupture,” and “endovascular,” covering publications from January 2000 to March 2024. Eligible articles included case reports, cohort studies, reviews, and consensus guidelines relevant to the epidemiology, diagnosis, and management of visceral arterial pseudoaneurysms (VAPAs). Articles were selected based on the clinical relevance and completeness of the anatomical or procedural detail described. The review was organized by embryological arterial origin (foregut, midgut, and hindgut) to reflect patterns in etiology, presentation, and intervention strategy across vascular territories. For each vascular territory, we have included illustrative case reports and representative radiological images to highlight clinically significant variations in presentation, diagnosis, and management. To minimize redundancy and support vessel-specific discussion, Table 1 summarizes embolization strategies based on aneurysm morphology, anatomical territory, and technical considerations.

## 2. Foregut

### 2.1. Coeliac Artery

#### 2.1.1. Epidemiology

Celiac artery aneurysms (CAAs) (including pseudoaneurysms) are the fourth most common aneurysm, constituting only 4% of visceral artery aneurysms [16]. The celiac trunk is the least commonly involved site for pseudoaneurysm formation as a result of chronic pancreatitis [73]. Reports of spontaneous or idiopathic celiac artery pseudoaneurysm formation or dissection cases are rare (34 cases in the 53 years from 1959) [74].

#### 2.1.2. Etiology

The causes of CAAs include infectious diseases, atherosclerosis, trauma, congenital conditions—such as median arcuate ligament syndrome—descending aortic dissection, spontaneous isolated celiac artery dissection [26], trauma [75], and iatrogenic causes [76,77,78].

#### 2.1.3. Presentation

The reported symptomatic presentations of celiac arterial pseudoaneurysms (CAPAs) are abdominal pain, vomiting, fatigue, and lightheadedness [20,26,48,50,74]. Spontaneous celiac artery dissection also may present with epigastric abdominal pain, especially in relation to a background of uncontrolled hypertension or blunt trauma.

On physical examination, epigastric tenderness may be present [74]. A CAPA can result in life-threatening hemorrhage due to rupture, and a patient may manifest clinical features of hemorrhagic shock [48].

#### 2.1.4. Diagnosis

Diagnosis of CAPAs is often achieved through contrast-enhanced CT [26,50,77]. DSA can aid in visualizing stenosis and the presence of thrombosis by retrograde filling of the coeliac artery via the gastroduodenal artery [48]. Multidetector CT angiography scans have also been used in its detection [20].

Laboratory findings of anemia, as well as low hematocrit and red blood cell counts, have been reported [28,48]. Other findings include elevated aspartate aminotransferase (AST), alanine aminotransferase (ALT), and bilirubin levels [78].

#### 2.1.5. Management

CAAs are commonly fusiform and located in the distal third of the celiac artery [12]. In cases where the origin of the celiac artery is involved, the use of endovascular techniques can be challenging due to the limited space for a seal zone that is required for a stent graft, as well as proximally for coil embolization [77], and is thus not technically feasible and not recommended [30]. In the case of high-grade stenosis at the origin, a percutaneous TAE can be employed in a retrograde manner via the superior mesenteric, common hepatic, and gastroduodenal arteries [48]. It is important to ensure adequate perfusion of organs supplied by main branches of the celiac artery (left gastric, common hepatic, and splenic arteries) to prevent end-organ ischemic damage [48]. The blood flow between celiac and superior mesenteric anastomosis, via the superior pancreaticoduodenal and inferior pancreaticoduodenal arcades, respectively, alleviates some of this concern [48]. A short celiac trunk may not allow for sufficiently stable embolization with coils, making a liquid embolic agent, such as a mixture of N-butyl-2-cyanoacrylate and ethiodized oil, more suitable [48]. Isolated spontaneous celiac dissections without serious complications have been found to resolve with conservative medical therapy (anticoagulation and/or antiplatelet therapy) alone [26].

### 2.2. Left Gastric Artery

#### 2.2.1. Epidemiology

Gastric artery pseudoaneurysms (GAPs) are extremely uncommon, accounting for less than 4% of cases [79,80], of which left gastric artery pseudoaneurysms (LGAPs) are the least common [81,82,83]. They are associated with a high rate of rupture and up to 70% mortality risk [84].

#### 2.2.2. Etiology

LGAPs usually occur after liver transplantation, pancreaticoduodenectomy, cholecystectomy, and bariatric procedures such as gastric bypass and sleeve gastrectomy [83]. Figure 3 shows an LGAP in a patient with blunt abdominal trauma secondary to a road accident. Gastric artery pseudoaneurysms are responsible for 75% of cases of upper gastrointestinal bleeding following sleeve gastrectomy [85]. They can also occur due to acute or chronic pancreatitis [81].

#### 2.2.3. Presentation

LGAPs are frequently asymptomatic [39], but symptomatic patients may experience diffuse abdominal pain [81,83]. Signs of bleeding and unstable vitals in the context of chronic pancreatitis warrant high clinical suspicion of an VAPA, including an LGAP. Patients may also present with hematemesis and melena. A delayed diagnosis can also result in presentation with upper gastrointestinal or spontaneous intraperitoneal hemorrhaging, as well as hemorrhagic shock [81,83].

#### 2.2.4. Diagnosis

Contrast-enhanced CT is especially advantageous in individualized pre-procedural planning specific to patients’ celiac axis anatomy [81]. Bedside ultrasounds can reveal a hypoechoic lesion around the lesser curvature of the stomach, as well as detect cholecystitis and pancreatitis [39]. An endoscopic ultrasound with Doppler functionality, due to its ability to examine the gastric wall, particularly in the case of intramural gastric aneurysms which can present as a submucosal tumor-like swelling during endoscopy, has been able to effectively diagnose submucosal gastric pseudoaneurysms [81]. In the case where diagnosis remains unclear after US, CT, and EUS, MRI can also be considered for the detection of LGAPs [82]. Attempts to perform a biopsy must be avoided during the endoscopy if a pseudoaneurysm is suspected.

Laboratory findings include anemia and low hematocrit percentage. Elevated amylase can be detected in acute pancreatitis patients. Uremia, elevated CRP levels, elevated D-Dimer levels, and a low anion gap can be found in the context of acute pancreatitis and cholecystitis [39,81,83].

#### 2.2.5. Management

The management of LGAPs is aligned with the general approach to the management of VAPAs outlined in the introduction, with a transarterial embolization (TAE) being the intervention of choice in a stable patient and surgical options such as arterial ligation and partial gastrectomy being preferred in the case of a rupture [81].

### 2.3. Esophageal Artery

Left gastric artery supply esophagus via esophageal arteries. Esophageal artery pseudoaneurysms (EAPs) are extremely rare, with only one reported case where Takayasu’s arteritis was the etiology [86]. As the esophagus occupies neck, thorax, and abdominal territories, the symptoms depend on the location of pathology. An acute onset of upper abdominal and back pain was reported. A left subclavian bruit was also detected. Apart from contrast-enhanced abdominal CT and CTA, an esophagogastroduodenoscopy is a useful investigation in which an external compression in the lower gastric body without gastroesophageal mucosal hemorrhage can be visualized. In terms of laboratory findings, a slightly elevated white cell count has been reported [86]. Their management is aligned with the general approach to the management of VAPAs outlined in the introduction, with no special considerations reported.

### 2.4. Splenic Artery

#### 2.4.1. Epidemiology

Splenic artery aneurysms (SAAs) constitute 50–75% of visceral artery aneurysms [58]. Splenic artery pseudoaneurysms (SAPAs) can have a rupture risk of up to 37% and mortality rate of almost 90% if not managed [38]. SAAs are four times more common in women but approximately three times more likely to rupture in men [87].

#### 2.4.2. Etiology

Pancreatitis (acute or chronic, but most commonly chronic) is the primary cause of SAPAs [53,88]. Other causes include non-iatrogenic trauma, which is increasingly recognized as a major contributor, particularly in the setting of blunt abdominal trauma involving the spleen. The splenic artery courses along the length of pancreas parenchyma and hence is the most susceptible in the context of necrotizing pancreatitis, with it being affected in 36% of cases [89]. Trauma secondary to falls or motor vehicle accidents and iatrogenic lesions secondary to hepatobiliary and pancreatic procedures are other significant causes [90]. Figure 4 shows an SAPA in a patient with blunt abdominal trauma due to road accident. Additional etiology includes infection, inflammatory processes (other than pancreatitis), peptic ulcer disease, and atherosclerosis [1,91,92].

#### 2.4.3. Presentation

SAPAs, especially smaller ones (<5 cm), are frequently asymptomatic and found incidentally [88,93]. Giant ones in particular can present as epigastric pain, vomiting, gastrointestinal bleeding, or hemorrhagic shock [93,94]. The most common presenting symptom is abdominal pain followed by hematochezia or melena and hemosuccus pancreaticus [94]. Other presentations include hematemesis, nausea, chest pain, back or flank pain, a feeling of fullness, and loss of appetite [38,88,94].

Patients often present with symptoms of rupture and consequent intraabdominal hemorrhaging and hemodynamic instability [1]. Hemorrhages most commonly occur in the pancreatic duct, followed by the stomach, peritoneal cavity, and colon [94]. Giant SAAs (≥5 cm) can be detected as a pulsatile mass in the upper-left quadrant or epigastrium [95]. Delays in diagnosing SAA are reported due to diagnostic confusion with pancreatic pseudocyst [92].

#### 2.4.4. Diagnosis

CT is the most commonly used modality in diagnosis, allowing for the visualization of anatomical details, complications, and etiological clues, which most frequently point to pancreatitis. It is difficult to differentiate between a pseudocyst and an SAPA in a non-contrast CT, and subsequent endoscopic intervention of a pseudocyst would lead to substantial hemorrhage, making a contrast-enhanced CT especially important [53]. It can further help differentiate SAPAs from pancreatic tumors, solid epithelial tumors, and gastric leiomyomas [88].

Laboratory findings of SAPAs include elevated serum amylase, serum lipase, leukocytosis, and anemia [96,97,98,99]. Blood glucose levels can also be elevated in the setting of acute pancreatitis [100].

#### 2.4.5. Management

According to the clinical practice guidelines of the Society for Vascular Surgery on the management of visceral aneurysms, all non-ruptured SAPAs of any size in patients of acceptable risk should be treated due to the possibility of rupture (Grade 1B) [101]. Open surgery, endovascular techniques (e.g., transcatheter embolization, percutaneous injection, and endovascular stent grafts) and laparoscopic excision are common management options [88]. The management of SAPAs is dependent on their size, location, dimensions, clinical severity, complications, and operative risks. A similar approach to management is applied to SAPAs and true splenic artery aneurysms [88].

Aneurysmectomy (with proximal and distal ligation of the artery without splenectomy) is indicated in aneurysms located in the proximal (hilar or intrasplenic) or middle third of the splenic artery, as well as in elongated and tortuous aneurysms. Resection with splenectomy with or without partial parietal pancreatectomy is indicated in aneurysms located in the distal third or two-thirds giant aneurysm and where dense strictures are present [38,58]. Endovascular stenting allows for preservation of the flow of the native splenic artery and minimizes splenic infarction and abscess complications that occur with coil embolization. It is suitable for an aneurysm in a proximal, less tortuous part of the vessel [102]. Transcatheter embolization is suitable when stenting is limited by tortuosity or decreased dimensions of the vessel, as well as for patients with surgical technical difficulty or at increased operative risk [103]. Percutaneous injection is indicated when embolization fails or is unsuitable [104,105].

### 2.5. Left Gastroepiploic Artery

Gastroepiploic artery aneurysms (GEAAs) are rare, constituting only 3.5% of visceral aneurysms [17], of which approximately 50% are gastroepiploic artery pseudoaneurysms (GEAPs) [57]. GEAAs have an incidence of about 0.1–0.4% [37]. A high rate of rupture at 90% [49,57] makes their prompt diagnosis and management crucial. GEAPs are chiefly attributable to abdominal trauma and less frequently to chronic pancreatitis, chronic cholecystitis, infection, iatrogenic trauma, systemic vasculitis, segmental arterial mediolysis (SAM), and cancer [17,37,49,106]. Reported left GEPA cases have been attributed to SAM and pancreatic pseudocysts due to pancreatic cancer-associated pancreatitis [107,108]. Unruptured aneurysms are frequently asymptomatic [49]. Patients present most commonly with abdominal pain or discomfort [17,108,109]. Ruptured GEAAs can present additionally with unconsciousness and shock due to intraperitoneal hemorrhaging [37,110]. They can also manifest as gradually increasing anemia with melena and hematemesis due to bleeding into the gastrointestinal tract, particularly the duodenum [49]. Left GEAPs have reportedly presented with a triad of abdominal pain, hematemesis, and anemia [107,108]. Laboratory findings include anemia [37,49,107] an elevated white cell count [37], elevated serum amylase, and elevated serum lipase [110]. Abdominal ultrasonography is non-specific but a useful baseline investigation which can be enhanced with color Doppler to enable detection of GEAPs with the yin-yang sign [110]. A sentinel clot sign, with dilation and stenosis of the left GEAP along with a potential hematoma, can also be observed via CT [108]. The management of GEAAs is aligned with the general approach to the management of VAPAs outlined in the introduction. Surgery (open surgery in unstable patients and laparoscopic surgery in stable patients) and TAE (in stable patients with high surgical risk) are indicated in the case of rupture [37]. Additionally, image-guided percutaneous embolization can be performed, where vessel tortuosity limits endovascular embolization [49].

### 2.6. Short Gastric Artery

Short gastric artery pseudoaneurysms are rare [111]. They can be caused by pancreatitis [111,112] and iatrogenic causes such as a gastrostomy placement [113]. Reported symptoms include hematemesis [111] and epigastric pain, and signs include anemia [112]. Diagnosis can be made through contrast-enhanced CT [111,112]. Endovascular treatment of a short gastric artery pseudoaneurysm requires the advancement of a microcatheter beyond the splenic artery, which can be challenging due to the tortuosity of the splenic artery [111]. A distal access catheter is useful in such cases, allowing for the deep advancement and successful embolization of aneurysms beyond a tortuous artery. However, a downside is that it requires a guiding sheath which is often soft [111]. Other reported management options include a coagulation of the pseudoaneurysm by percutaneous CT-guided thrombin injection, which can be considered an alternative to TAE when the pseudoaneurysm cannot be visualized on selective arteriography [112].

### 2.7. Common Hepatic Artery

#### 2.7.1. Epidemiology

Common hepatic artery pseudoaneurysms are extremely rare [21]. The incidence of hepatic artery aneurysms is uncommon at 0.002, with approximately 50% being hepatic artery pseudoaneurysms [21,114].

#### 2.7.2. Etiology

Reported etiologies include most commonly iatrogenic such as following total gastrectomy and upper abdominal surgery. Most iatrogenic causes have been reported to occur after biliary and pancreatic surgery. Other etiologies include infections and trauma [21].

#### 2.7.3. Presentation

Some patients may be asymptomatic [21,115,116]. Reported symptoms include massive gastrointestinal bleeding, melena, hematemesis, epigastric pain, non-specific abdominal pain, hematochezia, fatigue, nausea, and back pain [21,115,117].

Reported signs include signs suggestive of hypovolemic shock—including pallor, cold, and clammy peripheries—low blood pressure, tachycardia, and low oxygen saturation. Other signs reported are mild epigastric tenderness without peritonitis or organomegaly [21].

#### 2.7.4. Diagnosis

Investigations for common hepatic artery pseudoaneurysms are in line with the general investigations for VAPAs, with CTA being the recommended modality [118] alongside the additional use of upper GI endoscopy, as reported in the detection of thrombi or hemorrhages at the site of a jejunojejunostomy [21].

Laboratory findings include anemia, high blood sugar level, elevated bilirubin, low total protein and albumin, elevated platelets, and elevated ALT and AST [21,116].

#### 2.7.5. Management

Management options include arterial embolization, endovascular stent–graft repair (Figure 5) and open surgical repair (e.g., open reconstruction and aneurysmorrhaphy in the form of patch and ligation) aligned with those for hepatic artery aneurysms and VAPAs in general [101,116,118]. Specific considerations involve the endovascular embolization of the supplying vessel proximal to the hepatic artery pseudoaneurysm as the first-line approach [119], where perfusion to the liver can be preserved anatomically, with coil embolization being indicated in intrahepatic aneurysms and resection of the liver lobe being indicated in large aneurysms to reduce the risk of substantial liver necrosis [116]. Common hepatic artery aneurysms that are ruptured and in patients with fibromuscular dysplasia or polyarteritis nodosa require open surgical repair, with open surgical ligation recommended for the former and endoaneurysmorrhaphy for the latter [116].

### 2.8. Proper Hepatic Artery

Extrahepatic pseudoaneurysms are very uncommon and mostly present in a delayed manner [120,121]. About 1 in 5 splanchnic artery aneurysms involve the hepatic artery with a 2:1 predominance in males [122,123,124]. The most common cause is iatrogenic [122] after pancreaticobiliary surgery. Other reported etiologies include blunt hepatic trauma, infections, and penetrating trauma [120]. Pseudoaneurysms of the hepatic artery proper can be asymptomatic if unruptured or symptomatic [116,120]. The reported symptoms include epigastric pain, which are symptoms suggestive of obstructive jaundice [3,122] and gastrointestinal bleeding [125]. The reported signs include cardiac arrest from hemorrhagic shock, hypotension, and bradycardia [120]. Investigations for proper hepatic artery aneurysms are aligned with those for common hepatic artery aneurysms and VAPAs in general. Laboratory results can show mixed metabolic and respiratory acidosis from blood gases, anemia, and elevated levels of AST, ALT, platelets, and white cells [116,120,122]. Management options include resection with arterial reconstruction [122], endovascular treatment [126] and stent grafting [116,126,127], and open surgical repair with primary anastomosis [116,122]. Open surgical repair is reportedly the therapeutic regimen that has better efficacy and is usually chosen for cases of proper hepatic branch pseudoaneurysms, as it lowers the risk of hepatic ischemia, biliary complications, and abscess formation [116].

### 2.9. Right Gastric Artery

Isolated right gastric artery pseudoaneurysms (RGAPs) are a very rare complication of pancreatitis [55,128] but entail a potentially lethal complication of pancreatitis [55]. Reported etiologies include pancreatitis [129,130]. RGAPs may be asymptomatic or symptomatic [130]. Reported symptoms include severe asthenia and mild dyspnea, and signs include dependent oedema and intermittent melena [55]. Investigations for RGAPs are in line with those for VAPAs in general, with contrast-enhanced CTA and Doppler ultrasound reported to have been used for diagnosis [55,129,130]. Laboratory findings include anemia, low red cell count, and low hematocrit percentage [55]. Additionally elevated levels of WCC, amylase, lipase, C-reactive protein (CRP), procalcitonin, AST, ALT, total bilirubin, and conjugated bilirubin have been reported in the context of acute and chronic pancreatitis have been reported [55]. The management approach to RGAPs follows that of VAPAs in general, with no unique considerations reported. Both endovascular and surgical options can be used for management. Endovascular techniques include selective catheterization and embolization, percutaneous embolization, and vascular angioembolization [55,129,130].

### 2.10. Right and Left Hepatic Arteries

#### 2.10.1. Epidemiology

Hepatic artery pseudoaneurysms are uncommon [116], and pseudoaneurysms account for half of all hepatic artery aneurysms [6,116]. In a study by Khalil et al., the origin of the pseudoaneurysm as the right hepatic artery was seen in about 66% of the cases and from the cystic artery stump from 16.1% of the cases [131], which is a similar report to what Bulut et al. [132] reported, where the origin was from the right hepatic artery in two-thirds of the cases. The rupture of a hepatic artery pseudoaneurysm has been associated with a mortality rate of 21–43% [133,134,135,136,137]; hence, timely diagnosis is crucial.

#### 2.10.2. Etiology

The reported etiologies of hepatic artery pseudoaneurysms include iatrogenic injuries [116,131,138] such as following Whipple’s operation [116], via penetrating or blunt liver trauma [57,101,116,131,139], due to hepatobiliary infections such as chronic cholecystitis and acute cholangitis, and as a result of underlying malignancy [116]. Iatrogenic injury has been reported to be among the most common causes of hepatic artery pseudoaneurysms, especially after laparoscopic cholecystectomy [131,132,133,140,141]. In some reports, hepatic artery pseudoaneurysms occurred due to the contact with the proximal edge of a self-expandable metal stent used in the treatment of biliary obstruction due to malignancy [142,143].

#### 2.10.3. Presentation

Some pseudoaneurysms may be asymptomatic [132]. The reported symptoms most commonly include right hypochondriac pain [131]. Other reported symptoms include epigastric pain, abdominal pain, melena, nausea, and vomiting [116,131,133].

Reported signs include slight pressure pain in the upper abdomen with no rebound pain [116], right abdominal tenderness with abdominal guarding, hemorrhagic shock, overt bleeding in the drains, obstructive jaundice, hypotension, upper gastrointestinal bleeding, and pallor [116,131,133]. Hemobilia in patients with biliary intervention requires a careful assessment to check for hepatic artery pseudoaneurysm (Figure 6).

#### 2.10.4. Diagnosis

CTA can be used for diagnosis and is useful in detecting the size and the origin of the pseudoaneurysm. Other investigations include CT scan of the abdomen with intravenous contrast and DSA [116,131,133].

Reported laboratory findings include anemia, low hematocrit levels, and raised lactate levels [133].

#### 2.10.5. Management

Left and right hepatic arteries: Management options include open surgical repair and endovascular approaches through interventional radiology by insertion of an arterial stent or coiling at the neck of the aneurysm or through coil TAE [116,119,131,133].

Approaching the pseudoaneurysm in the presence of a thrombosed proximal vessel, such as tortuous hepatic arteries [133], can pose challenges to management with TAE, requiring further attempts at embolization. There have been reports on the successful management of hepatic pseudoaneurysms by injecting thrombin directly into it [133,141,144,145]; however, embolization using this technique may be non-selective, resulting in potentially serious complications such as liver and bowel infarctions—and injecting small aliquots of thrombin with real-time ultrasound and Doppler guidance has been posited to reduce this risk [141,144].

More complex surgery has been reported to be required in cases where the aneurysm was in the main hepatic artery where reconstruction with the splenic artery was needed; in the case where reconstruction is not possible, right hepatectomy may be needed to avoid the occurrence of liver infarction and abscess [131].

According to the clinical practice guidelines of the Society for Vascular Surgery on the management of visceral aneurysms, all hepatic artery pseudoaneurysms, due to the high propensity of rupture and significant antecedent mortality and regardless of cause, should be repaired as soon as the diagnosis is made (Grade 1A) [101].

### 2.11. Cystic Artery

#### 2.11.1. Epidemiology

Cystic artery pseudoaneurysms (CAPs) are very rare [40,41]. The age range of previously reported cases of CAPs cases was 32 to 90 years, with a median age at diagnosis of 68 years [40,41]. Most patients are elderly males [40]. They tend to enlarge and erode into the gallbladder and adjacent biliary tree, with approximately 45% bleeding into the biliary system (hemobilia) [42,146]. Figure 7 demonstrates a patient with CAP presenting with per rectal bleeding. Figure 8 demonstrates a patient with CAP and associated hemoperitoneum.

#### 2.11.2. Etiology

Most common etiologies include inflammation and trauma [40,41,42]. Reported etiologies include acute cholecystitis, xanthogranulomatous cholecystitis, cholelithiasis, gallstone disease, malignancy, iatrogenic such as biliary tract manipulation, and pancreatitis [40,41,42,147]. A well-documented iatrogenic cause of CAPs is postlaparoscopic cholecystectomy, where pseudoaneurysms typically involve the cystic artery stump or adjacent right hepatic artery branch. These may develop due to thermal or clip-related vascular injury during dissection of Calot’s triangle. Notably, the development of CAPs can be accelerated by patients’ comorbidities such as atherosclerosis, hypertension, bleeding disorders, and vasculitis [7,8,9,10,42,148]. Rarely, Median Arcuate Ligament Syndrome (MALS), a rare vascular disorder caused by an extrinsic compression of the celiac artery from the median arcuate ligament, is associated with pseudoaneurysm formation due to compression [149].

#### 2.11.3. Presentation

While some may be asymptomatic and detected incidentally on imaging, the most commonly reported symptom of cystic artery pseudoaneurysms is abdominal pain, followed by upper gastrointestinal bleeding and combined symptoms of Quincke’s Triad, which consists of jaundice, right upper quadrant pain, and upper gastrointestinal bleeding [40,41,42]. Other reported symptoms include fever and bilious vomiting [40,41,42].

Reported signs include signs suggestive of hypovolemic shock, pallor due to anemia, hypotension, tachycardia, right upper quadrant abdominal tenderness, melena, fresh per-rectal bleeding, and jaundice [40,42]. The duration of bleeding determines presentation with hemorrhagic shock in the case of massive acute bleeding and anemia, jaundice, pancreatitis, cholangitis, or cholecystitis in the case of more prolonged bleeding as a result of the biliary tree being obstructed by clots [40].

#### 2.11.4. Diagnosis

Investigations for CAPs are in line with the general investigations for VAPAs with no additional specifications [40,42]. Common laboratory findings are hyperbilirubinemia, anemia, leukocytosis, and increased levels of ALP, AST, GGT, ALT, CRP, total bilirubin, and conjugated bilirubin [40,41,42].

#### 2.11.5. Management

Definitive management of CAPs can be surgical with a cholecystectomy, endovascular via angiography and angioembolization, or a combination of both [40,41,42]. Temporary management options include percutaneous cholecystostomy used in patients reported to have ruptured cystic artery pseudoaneurysms in the setting of acute calculous cholecystitis [40].

### 2.12. Gastroduodenal Artery

#### 2.12.1. Epidemiology

Gastroduodenal artery pseudoaneurysms (GADPs) make up about 1.5% of reported visceral pseudoaneurysms [150]. The chance of rupture of GADPs is as high as 90% without correlation with size; hence, asymptomatic patients are also routinely embolized to prevent mortality if untreated [23,33,150].

#### 2.12.2. Etiology

Reported etiologies of GADPs include most commonly pancreatitis, especially chronic pancreatitis [5,23,150,151,152], as well as trauma [5,9,23], iatrogenic [153], vascular abnormalities [23,154], ethanol abuse [3,5,23], peptic ulcer disease [23], and as a result of postoperative pancreatic fistula after pancreaticoduodenectomy. Figure 9 shows a CT scan with a GDA pseudoaneurysm in a patient operated for laparoscopic subtotal gastrectomy with D2 lymphadenectomy for lesser curve gastric cancer. Figure 10 demonstrates a patient with GDA associated with biliary stenting. Rarely, spontaneous GADPs occur [23,155].

#### 2.12.3. Presentation

GADPs may be asymptomatic [5,150,152,153]. Reported symptoms include gastrointestinal bleeding [5,23,150,156], abdominal pain [5,23,150,152,156,157], compressive symptoms such as nausea [152,153] and vomiting [5,153], and diarrhea [5].

Reported signs include signs suggestive of obstructive jaundice [5,150,152,158], melena [5,150,157], hematemesis [5,153], epigastric tenderness [150] or presence of a pulsatile abdominal mass with or without bruit [5,153], and shock [153]. In patients who underwent pancreaticoduodenectomy, blood-stained drain effluent should be considered serious finding, and prompt attention is necessary to establish a diagnosis of underlying GADP.

#### 2.12.4. Diagnosis

Approaches to investigations for GADPs are aligned with that of VAPAs in general [5,23,150,152,157]. Laboratory findings include anemia; elevated ALT, AST, and total bilirubin raised lipase; and elevated CRP [150,152]. Other findings include elevated WCC, urea, and amylase [23]. Elevated serum amylase or lipase levels may reflect underlying pancreatitis in patients with GDAP, but they are non-specific. Suspicion should be confirmed with imaging modalities such as CT angiography or DSA.

#### 2.12.5. Management

GADP is a known complication following pancreaticoduodenectomy, and thus, certain preventive maneuvers have been reported. Covering the stump with omental or falciform ligament tissue and tying the stump securely at least twice are common operative rituals. In addition, a sufficient length is kept from the origin in anticipation that if a pseudoaneurysm develops, there is sufficient space for coils to be placed within the vessel stump. Lastly, metal clips are applied in addition to suture ligatures in anticipation that this aids in the detection of the gastroduodenal artery stump that becomes easier upon fluoroscopy. Management options include surgical (revascularization, vessel ligature, and aneurysmal sac exclusion) and endovascular interventions (coil embolization and stent placement) (Figure 7) [5,23,150,152,153] depending on the symptoms, location of the pseudoaneurysm, patient’s clinical status, and the risk of organ ischemia after intervention [5] and anatomy [152].

### 2.13. Right Gastroepiploic Artery

#### 2.13.1. Epidemiology

Right GEAPs are rare and potentially life threatening due to risk of rupture [17]. They are three times more common in men than in women, occurring at an average age of 65 years and account for fewer than 4% of all aneurysms involving the splanchnic arteries [159].

#### 2.13.2. Etiology

Reported causes include iatrogenic including endoscopic ultrasonography [37], vasculitis such as Churg–Strauss Syndrome [160] and pancreatitis [17]. Figure 11 shows a clinical profile and management of spontaneous RGEP in an elderly female patient.

#### 2.13.3. Presentation

Presentation may be asymptomatic. Reported symptoms include abdominal pain weakness and hematochezia. Reported signs include ascites, superior epigastric abdominal tenderness to palpation with guarding, and hemorrhagic shock if ruptured [160,161,162].

#### 2.13.4. Diagnosis

GEAPs share a similar approach to investigations as VAPAs in general, with diagnoses reportedly having been made on CT and catheter angiography [160,161]. Laboratory findings of anemia [17] leukocytosis, and elevated CRP levels [160] have been reported.

#### 2.13.5. Management

Management options include transcatheter arterial coil embolization and surgery such as open en bloc resection of the aneurysm, a portion of the stomach, and a portion of the transverse colon, especially with ruptured right GEAPs [17,160,161,162].

### 2.14. Superior Pancreaticoduodenal Artery

#### 2.14.1. Epidemiology

Superior pancreaticoduodenal artery pseudoaneurysms (SPAPs) are uncommon [130].

#### 2.14.2. Etiology

Reported causes associated with SPAPs include iatrogenic causes, specifically from metal stents, resulting in a posterior SPAP [163] and anterior superior pancreaticoduodenal artery pseudoaneurysm after distal pancreatectomy with en bloc celiac axis resection [164], acute pancreatitis [130], and chronic pancreatitis [165].

#### 2.14.3. Presentation

SPAPs may be asymptomatic [130,166]. They have also been reported to present with symptoms mimicking a cystic neoplasm, including poor appetite, unintentional weight loss, and intermittent diarrhea [166]. The reported signs include hemodynamic collapse from severe bleeding [166]. Figure 12 shows an SPAP following a pancreatic lipomatous tumor resection complicated by a disconnected pancreatic duct and managed with glue embolization in a male patient.

#### 2.14.4. Diagnosis

CT scan, CT angiogram, endoscopic ultrasound (EUS), and EUS–fine needle aspiration with Doppler US can be used to diagnose SPAPs [130,163,164,165]. Laboratory findings of anemia [163], low hematocrit levels [165], neutrophilic leukocytosis, and increased amylases, lipases, CRP, and procalcitonin levels have been reported [130].

#### 2.14.5. Management

Management options include angiography with coil embolization and laparotomy, ligation, and excision of the pseudoaneurysm should angiographic embolization be unsuccessful [130,163,164,165].

## 3. Midgut

### 3.1. Superior Mesenteric Artery

#### 3.1.1. Epidemiology

Superior mesenteric artery pseudoaneurysms (SMAPs) are amongst the least frequently reported VAPAs, with superior mesenteric artery and its branches of pseudoaneurysms accounting for about 5.5% of all VAPAs [167] and having an overall low incidence of 0.01% [167,168]. However, given the relatively high mortality rates associated with SMAPs [24,168] and high risk of rupture of 38% to 50% higher than other visceral aneurysms [169], it is vital to understand its clinical presentations.

#### 3.1.2. Etiology

The etiologies of SMAPs include infections such as infective endocarditis, inflammatory processes—particularly pancreatitis—iatrogenic injury, non-iatrogenic traumatic injury, and other disease processes such as atherosclerosis and uncontrolled hypertension [91,168,170]. Notably, pancreatitis and trauma are the most common causes of SMA pseudoaneurysms [24,167,169]. Additionally, the SMA is the most common location for a visceral mycotic pseudoaneurysm, comprising 88% of visceral mycotic pseudoaneurysms [168].

#### 3.1.3. Presentation

SMAPs may be asymptomatic and found incidentally [171]. Abdominal pain is the most commonly noted symptom of SMAPs [168,169,170]. Other symptoms include vomiting, fever, diarrhea, or visceral pain upon eating (abdominal colic syndrome) [169,171].

Signs associated with SMAPs include intraabdominal bleeding, mesenteric hematoma, intraluminal rupture with gastrointestinal bleeding, pulsatile masses, and sudden hypotension [169,171]. Due to its blood supply to the small intestine, SMAPs have potential to cause mesenteric ischemia.

#### 3.1.4. Diagnosis

Investigations for SMAPs are aligned with those of VAPAs in general [91,167,168,170,172]. Notably, mesenteric angiography with or without digital subtraction is more sensitive for diagnosing SMAPs MRI, CT, and abdominal ultrasound with CT angiograms as the modality of choice [168]. Furthermore, positron emission tomography (PET)/CT is useful to investigate complications of SMAPs such as septic embolic events [168,169,173,174].

Additionally, neutrophil elevations have been found in mesenteric artery pseudoaneurysms secondary to immune-mediated vasculitis such as polyarteritis nodosa [175], pointing toward the possibility of using the neutrophil count as a potential marker for pseudoaneurysms secondary to immune-mediated etiologies.

#### 3.1.5. Management

Overall, the management of SMAPs is aligned with that of VAPAs in general [25,91,167,176,177]. Notably, coil embolization is the primary treatment modality [176] for two-thirds of superior mesenteric artery aneurysms and pseudoaneurysms, with a reported success rate of 89% [169]. Catheter-based endovascular techniques are particularly useful for high-risk patients and patients with anatomically challenging lesions [168]. One of the important determinants of the outcome is mesenteric vascularity and bowel viability. In situations of doubt, indocyanine green dye study can be used to assess mucosal perfusion [178]. In situations of doubtful viability, an open abdomen with a planned relook laparotomy should be considered [179].

Management differs based on special situations. Particularly, ligation without reconstruction [168] is commonly performed in emergency settings. In the treatment of SMAPs due to infective endocarditis, placing a covered stent with a full course of antibiotics before and after surgery could likely be a successful alternative to open surgery, with effective antibiotic control being crucial in management [169].

### 3.2. Inferior Mesenteric Artery

#### 3.2.1. Epidemiology

Inferior pancreaticoduodenal artery pseudoaneurysms (IPAPs) are extremely uncommon [180,181,182], accounting for less than 10% of VAPAs [180]. Morbidity and mortality have been reported to be greater than 25% [180].

#### 3.2.2. Etiology

The most common causes of IPAPs are iatrogenic or following trauma [180]. Other etiologies include pancreatitis [183,184,185] and, uncommonly, severe cholecystitis [186], septic emboli, and laparoscopic cholecystectomy [187,188,189].

#### 3.2.3. Presentation

The symptoms include abdominal pain, melena or hematochezia, hematemesis, biliary colic, jaundice, hemodynamic instability, and shock [180]. Reported signs include a soft and mildly tender abdomen in the epigastric region, with no evidence of peritonism [186].

#### 3.2.4. Diagnosis

Like VAPAs in general, US, Doppler US, CT, and angiography are diagnostic modalities that can diagnose pancreatic pseudoaneurysms. Abdominal contrast-enhanced CT scanning is usually sufficient for appropriate identification of pancreatic pseudoaneurysms; however, it should be angiographically confirmed to localize the bleeding pseudoaneurysm [186,190]. DSA can also be performed together with endovascular coil embolization to confirm the location of the pseudoaneurysm [184].

The reported laboratory results in patients with inferior pancreaticoduodenal artery pseudoaneurysms include those suggestive of acute kidney injury, including significantly raised urea and creatinine levels [186]. Additionally, low hemoglobin levels [186], leukocytosis, and elevated CRP and fibrinogen levels have been reported [184]. Biochemistry results suggestive of etiologies such as acute on chronic pancreatitis have shown high amylase and lipase levels [184].

#### 3.2.5. Management

Ligation of the vessel that feeds the pseudoaneurysm is the most appropriate treatment option in management [191]. Currently, angioembolization has been considered the most appropriate treatment option for the treatment of visceral pseudoaneurysms associated with pancreatitis due to its ability to achieve hemostasis [192,193]. Other options include percutaneous thrombin injections, which is cheaper but more invasive. Thrombin injection is suitable for pseudoaneurysms with a narrow neck, while angioembolization is recommended for those with a wide neck [193,194].

There is a lack of data regarding endovascular stent graft placement in the IPAP [195]. There have been reported cases where minimally invasive endovascular treatment such as the sandwich technique were performed. The sandwich technique involves both inflow and outflow of the pseudoaneurysm being embolized with coils [184]. When all other options are exhausted, pancreatic resection along with the pseudoaneurysm can be done. However, this is particularly avoided in emergency settings, as it has been associated with a high mortality rate [191].

### 3.3. Jejunal Artery

#### 3.3.1. Epidemiology

Jejunal artery pseudoaneurysms are extremely uncommon [196,197,198,199], representing less than 1% of all visceral artery pseudoaneurysms [198] and affecting mostly men [4].

#### 3.3.2. Etiology

The etiologies reported include pancreatitis [200], Crohn’s disease [201], infections such as intestinal tuberculosis [196,202], endocarditis [198], jejunal arterial fistula [203], and gastroepiploic artery operation history [199,204,205,206]. Notably, the main etiology for jejunal pseudoaneurysms is bacterial infection, with Streptococcus being the most frequent pathogen, especially in patients with endocarditis [159,198,206].

#### 3.3.3. Presentation

Patients with jejunal artery pseudoaneurysms may be asymptomatic or present with gastrointestinal hemorrhage [196].

#### 3.3.4. Diagnosis

Apart from CTA [200], Doppler ultrasound can be used to survey for hypoechoic cystic lesions with proximity to an artery and evaluate active blood flow [199], and computed tomography enterography (CTE) can be used to assess extent [201].

Similar to other midgut artery pseudoaneurysms, reported laboratory results in jejunal artery pseudoaneurysm include low hemoglobin levels showing severe anemia and elevated amylase and lipase in pseudoaneurysms secondary to pancreatitis [199].

#### 3.3.5. Management

Angiographic arterial embolization can be performed as first-line treatment [196,200]. However, in mycotic jejunal artery pseudoaneurysms, endovascular embolization is not ideal due to the possibility of bacterial proliferation. In such cases, open repair is most often considered definitive treatment [101,198].

Placement of a stent graft may be problematic in jejunal pseudoaneurysms due to the possibility of occlusion of other normal vascular branches next to the pseudoaneurysm by extension of the stent graft, which greatly increases the risk of intestinal ischemia. A stent graft can also be complicated by occlusion, and it usually requires prolonged postprocedure anticoagulation therapy. Alternatively, ultrasound or CT-guided direct percutaneous puncture of the pseudoaneurysm followed by the injection of cyanoacrylate or thrombin can be performed, but this may be difficult due to localization of the pseudoaneurysms and the interposition of intestinal loops [197,199].

### 3.4. Ileal Artery

Ileal artery pseudoaneurysms are extremely rare [207]. Apart from the general causes of VAPAs, reported etiologies for ileal artery pseudoaneurysms include posttrauma [208] and iatrogenic outcomes [207]. Ileal artery pseudoaneurysms can present with abdominal pain [207] and gastrointestinal bleeding [209]. CT angiography is considered the gold standard for diagnosis and planning of therapeutic interventions [208]. Contrast-enhanced abdominal CT can also be used for diagnosis [207]. In an ileal artery pseudoaneurysm secondary to trauma reported in a child, the white blood cell (WBC) count was elevated, and the hemoglobin level was low [210]. The management of ileal artery aneurysms is likely to follow that of VAPAs in general, although there is limited available literature on the topic due to its rarity [207,211]. Surgical treatment is usually performed, although percutaneous embolization is also a safe intervention, with the use of coils with the sandwich technique being preferred [208]. Figure 13 demonstrates a young female patient with immunodeficiency disorder treated with small bowel resection complicated by ileal artery pseudoaneurysm treated with coil embolization.

### 3.5. Middle Colic Arteries

Middle colic artery pseudoaneurysms (MCAPs) are among the rarest VAPAs, accounting for approximately 6–8% of all VAPAs with an incidence of 0.01% [212]. The etiologies reported include acute pancreatitis [213], infection, iatrogenic, and trauma [212]. MCAPs can be asymptomatic [212] or present with the acute onset of epigastric discomfort, dizziness, diaphoresis [214], abdominal pain, nausea, and vomiting [212]. Presenting signs reported include hemorrhagic shock [214], massive bleeding [213], pulsatile mass or a bruit, and gastrointestinal bleeding [212]. MCAPs have been diagnosed using CT of the abdomen with contrast enhancement and angiography [212,213,214]. Other imaging modalities, including ultrasound and Doppler sonography, can also be used in diagnosis [212]. The reported laboratory results include low hemoglobin levels [213]. MCAPs can be managed either conservatively or invasively. Conservative management can be considered clinically stable patients with no evidence of rupture or ischemic bowel change [213]. Invasive management includes embolization with glue [212,213,214]. Continuous regional arterial infusion of a protease inhibitor and antibiotics [213], endovascular therapy in hemodynamically stable patients, and open surgical resection [212,214]. Figure 14 shows an elderly female patient with incidental MCAP diagnosed during the management of pancreatic neuroendocrine tumors.

### 3.6. Right Colic Arteries

Right colic artery pseudoaneurysms (RCAPs) are extremely rare [215,216]. General etiologies include trauma, infection, and inflammation [215]. The symptoms reported include right upper abdominal pain [215], epigastric pain, fever, and vomiting [216]. Signs suggestive of hemodynamic instability have been reported [215]. CT has reportedly been the mainstay for diagnosis of RCAPs [215,216]. In a reported case of hemorrhage secondary to a rupture of a right colic artery pseudoaneurysm, the laboratory results showed elevated CRP and INR levels [215]. The initial management options suggested include endovascular such as transcatheter arterial embolization [216], with surgery being considered only where infrastructure is lacking, when there is a failure of endovascular approach, or its risks outweigh the benefits. Surgical options include a laparotomy, excluding the pseudoaneurysm, ligating the culprit vessel, and addressing the end organ as suitable [215].

### 3.7. Superior Mesenteric Artery

#### 3.7.1. Epidemiology

Ileocolic artery pseudoaneurysms are extremely rare, with very few cases being reported [216,217].

#### 3.7.2. Etiology

Commonly reported etiologies include pancreatitis, with other less common etiologies being abdominal trauma, iatrogenic trauma, vasculitis, atheromatous causes, inflammation such as Crohn’s disease [218,219], and infection [217,220].

#### 3.7.3. Presentation

Ileocolic artery pseudoaneurysms can be asymptomatic or symptomatic [217,221], presenting with abdominal pain, lower intestinal bleeding, and melena [217,220,221]. Reported signs in patients presenting with ileocolic artery pseudoaneurysms include intra-abdominal hemorrhage and signs of hypovolemic shock [220,221].

#### 3.7.4. Diagnosis

Abdominal contrast-enhanced CT and angiography can successfully diagnose ileocolic artery pseudoaneurysms [217,218,219,220].

Similar to other ruptured VAPAs, reported ruptured ileocolic artery pseudoaneurysms showed leukocytosis and elevated CRP levels [218,220].

#### 3.7.5. Management

Interventional angiographic treatment is increasingly the preferred approach of management, since it is minimally invasive and has high success and low mortality rates [217,218,219,220]. Specifically, reports have used this approach to catheterize the iliac artery from the superior mesenteric artery, and angiography was performed to visualize the ileocolic artery pseudoaneurysm, with a microcatheter and microcoil being inserted for hemostasis [220]. Alternatively, surgical approaches can also be considered [217,220].

## 4. Hindgut

### 4.1. Inferior Mesenteric Artery

#### 4.1.1. Epidemiology

Inferior mesenteric artery (IMA) pseudoaneurysms are extremely rare compared to other visceral artery pseudoaneurysms [159,222], with an incidence of roughly 1% [28,223]. There has been only one report of inferior mesenteric sigmoid branch pseudoaneurysm in the literature [222], and only around 60 cases of inferior mesenteric pseudoaneurysm have been reported to date [223].

#### 4.1.2. Etiology

Reported etiologies most commonly include atherosclerosis, followed by mycotic, polyarteritis nodosa, dissecting hematoma, Takayasu’s disease, iatrogenic, aortitis, segmental mediolytic arteritis, tuberculosis, Behcet’s disease, neurofibromatosis type 1, and trauma [222]. There has been one case of traumatic superior rectal artery pseudoaneurysm [222,224] and one case of left colic artery pseudoaneurysm secondary to pancreatitis [222,225] reported.

#### 4.1.3. Presentation

Patients with IMA pseudoaneurysms have been reported to present with lower and upper gastrointestinal bleeding, diffuse abdominal pain, the inability to pass stools, and flatus. Reported signs include abdominal tenderness and obstipation [222].

#### 4.1.4. Diagnosis

CECT has been used in the diagnosis of IMA pseudoaneurysms [222]. In IMA pseudoaneurysms secondary to pancreatitis, elevated amylase and lipase levels have been noted [226].

#### 4.1.5. Management

IMA pseudoaneurysms have been managed with selective coil embolization, with endovascular treatment currently preferred due to its safety and less-invasive nature. Alternatively, surgical approaches may be considered [222].

### 4.2. Left Colic Artery

#### 4.2.1. Epidemiology

IMA pseudoaneurysms, including left colic artery pseudoaneurysms, are extremely rare [223].

#### 4.2.2. Etiology

Reported etiologies include pancreatitis [225] and iatrogenic causes such as postrenal biopsy due to its location close to the left kidney [36].

#### 4.2.3. Presentation

Reported symptoms include lower abdominal pain [36,223] and increasing swelling in the left flank region [36]. Reported signs include active duodenal bleeding, recurrent gastrointestinal bleeding [225], tachycardia, and low blood pressure [36].

#### 4.2.4. Diagnosis

Although reported cases showed that esophagoscopy could detect brisk duodenal bleeding and selective inferior mesenteric arteriography, including an IMA injection could be performed, diagnostic yield could improve using a three-vessel mesenteric arteriogram instead [225]. Other reports have shown successful diagnosis using selective IMA arteriography, CT scans of the abdomen and pelvis [223], and CT angiography, with the pseudoaneurysm better seen on the venous phase due to its narrow neck [36]. IMA DSA through the transfemoral route has also been used for diagnosis [36]. In a reported left colic artery pseudoaneurysm, significant laboratory results included hyponatremia, hypocalcemia, and elevated prothrombin time [223].

#### 4.2.5. Management

Left colic artery pseudoaneurysms can be managed with transcatheter arterial embolization and coil embolization [36,223,225].

### 4.3. Sigmoid Artery

#### 4.3.1. Epidemiology

IMA pseudoaneurysms are extremely rare compared to other visceral artery pseudoaneurysms [159,222], with an incidence of roughly 1% [28,223]. There has been only one report of inferior mesenteric sigmoid branch pseudoaneurysm reported in the literature [222].

#### 4.3.2. Etiology

Commonly reported etiologies include pancreatitis, abdominal trauma, vasculitis, and abdominal sepsis.

#### 4.3.3. Presentation

Reported symptoms and signs of sigmoid artery pseudoaneurysms include hematemesis and rectal bleeding [222].

#### 4.3.4. Diagnosis

CECT has been used for diagnosis of sigmoid artery pseudoaneurysms [222,227], with DSA being used to confirm a pseudoaneurysm arising from the sigmoid branch of IMA between the left colic and superior rectal branches [222].

#### 4.3.5. Management

Imaging-guided percutaneous puncture and embolization of sigmoid artery pseudoaneurysms has been used for management [227], with endovascular treatment being a safe and less-invasive treatment option for treatment [222].

### 4.4. Superior Rectal Artery

#### 4.4.1. Epidemiology

Visceral pseudoaneurysms are extremely uncommon, with only 1% of cases affecting the IMA and its branches [11,25,228,229]. The superior rectal artery is a terminal branch of the IMA in which bleeding pseudoaneurysms are seldom reported without a history of trauma or procedure [228]. Superior rectal artery pseudoaneurysms (SRAPs) are extremely rare, having been documented in the literature only seven times, and can be extremely dangerous [11].

#### 4.4.2. Etiology

Reported etiologies include trauma [224,230] and iatrogenic trauma [229].

#### 4.4.3. Presentation

SRAPs have been reported to present with lower abdominal pain, dizziness, severe hematochezia constipation, vague abdominal complaints, and pain [11,228,230].

Signs include lower gastrointestinal bleeding, pallor, tenderness to palpation in the left lower quadrant with active large-volume hematochezia, hypotension, tachycardia fever, retroperitoneal hematomas, and can progression to hemodynamic instability related to hypovolemia [11,25,228,229].

#### 4.4.4. Diagnosis

Abdominal CT angiogram has been used for diagnosis, showing focal wall thickening of the distal sigmoid colon and active contrast extravasation suggestive of an active hemorrhage. Interventional radiology performed visceral arteriography has also been used for diagnosis, demonstrating active extravasation associated with a pseudoaneurysm in a branch of the superior rectal artery [228]. Angiography can be used for both diagnostic and therapeutic purposes [11,224,229,230]. Significantly low hemoglobin levels have also been reported in cases of superior rectal artery pseudoaneurysms [224,228].

#### 4.4.5. Management

SRAPs have been managed using endovascular approaches such as coiling and injection of procoagulant substances to control bleeding from ruptured pseudoaneurysms that fail to settle spontaneously [228], as well as angiography followed by selective superior rectal artery embolization [11,224,230]. To reduce the area at risk for ischemia, the most distal site should be chosen for embolization [224,231,232], so super-selective embolization of the superior rectal artery theoretically reduces the risk of ischemic complications of sigmoid colon and rectum. While surgical management is possible, the superior rectal artery may be surgically difficult to access [224].

## 5. Limitations and Future Perspectives

### 5.1. Limitations

Due to the rarity of VAPAs, there is a limited amount of literature available specifically on VAPAs arising from the various visceral arterial branches. This review, therefore, relies significantly on case reports and case series which may be unique to the individual cases discussed and not generalizable across populations, particularly due to the anatomical variants in visceral arteries which may affect their management. Future research can focus on determining whether the epidemiology, etiologies, presentations, diagnostic modalities, and management options of VAPAs can be applied to populations at large. Another important consideration is the lack of robust long-term follow-up data across many case reports and series. Given the potential for recurrence, reintervention, or late ischemic complications following embolization, establishing standardized protocols for surveillance imaging and clinical review would enhance patient outcomes and evidence-based care planning.

### 5.2. Future Perspectives

#### 5.2.1. Investigations

With the shift toward artificial intelligence (AI) in medicine, the potential roles of AI in automated image segmentation, vessel mapping, the detection of subtle contrast extravasation, and risk prediction models should be explored to improve the diagnostic accuracy and speed of pseudoaneurysm detection, particularly in CT angiography and digital subtraction angiography [233].

The role of Indocyanine Green (ICG) can be pivotal during intraoperative assessment. In patients undergoing open or laparoscopic management of pseudoaneurysms, especially when bowel ischemia is suspected, ICG fluorescence angiography offers real-time visualization of visceral perfusion. It may help guide resection margins or assess tissue viability following embolization in ischemia-prone territories [234,235,236].

While this paper discusses reported blood test and serological results suggestive of the presence of pseudoaneurysms, the roles of novel serology biomarkers for the early diagnosis of ischemic or infarcted body viscera are potential areas of further study in order to detect and manage pseudoaneurysms efficiently in the future [237].

#### 5.2.2. Management

Currently, the reported indications of surgery for pseudoaneurysms do not primarily take into consideration transfusion requirements of patients, and future studies could consider possible indications of transfusion prior to surgery [238]. In the surgical management of pseudoaneurysms requiring splenectomy, adequate and timely initial immunization in both elective and emergency surgeries can be further explored to evaluate the optimal vaccination regimes [239]. To improve the understanding of VAPA incidence, etiology, and outcomes, a multicenter registry or global collaborative database should be developed. Such efforts would enable standardization of data collection, foster comparative research, and support evidence-based guideline development for this rare but high-risk condition. Recent multidisciplinary guidance, such as the Italian consensus by Pratesi et al., supports artery-specific diagnostic and treatment algorithms in the management of visceral and renal aneurysms [240].

#### 5.2.3. Ethical and Legal Implications

Global disparities in healthcare resources also influence the diagnostic and therapeutic approaches to VAPAs. In low- and middle-income countries, limited access to endovascular suites, advanced imaging, and trained interventionalists may necessitate greater reliance on open surgical interventions or delayed diagnosis, impacting outcomes. Given that pseudoaneurysms secondary to iatrogenic causes are not uncommon, it is important to consider potential ethico-legal implications of pseudoaneurysms occurring after planned elective surgical procedures such as bariatric surgery, especially since they can have life-threatening consequences [241].

## 6. Conclusions

To the best of our knowledge, this is the first comprehensive review on VAPAs. Given their high risk of rupture and associated morbidity and mortality, understanding their epidemiology, etiologies, presentations, diagnostic modalities, and management options is crucial. While diagnostic modalities are similar across the various VAPAs, there are differences in epidemiology, etiologies, presentation, and management that should be taken into account when assessing patients.

## Figures and Tables

**Figure 1 medicina-61-01312-f001:**
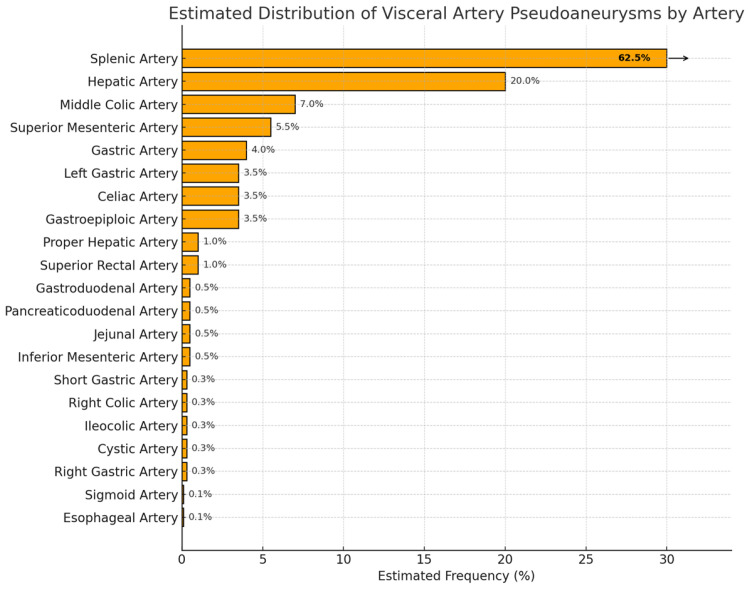
Estimated distribution of VAPA by type of artery.

**Figure 2 medicina-61-01312-f002:**
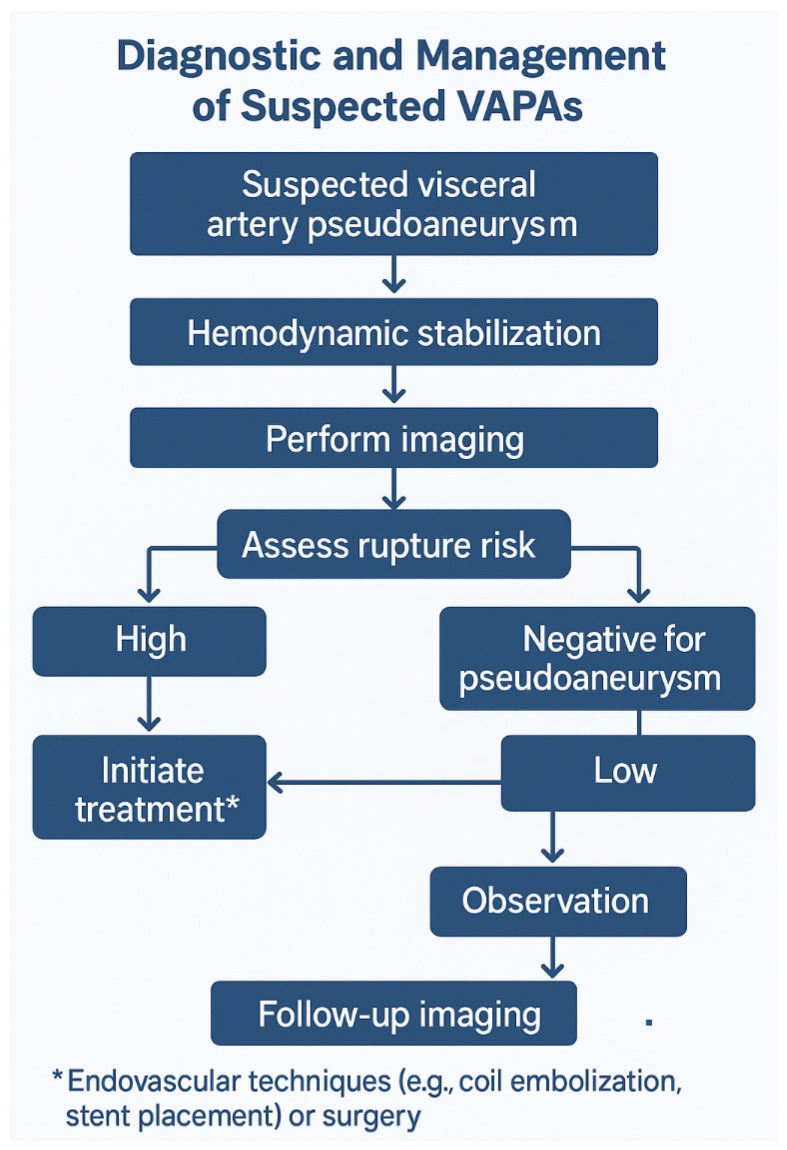
Flow chart showing diagnosis and management of suspected VAPAs.

**Figure 3 medicina-61-01312-f003:**
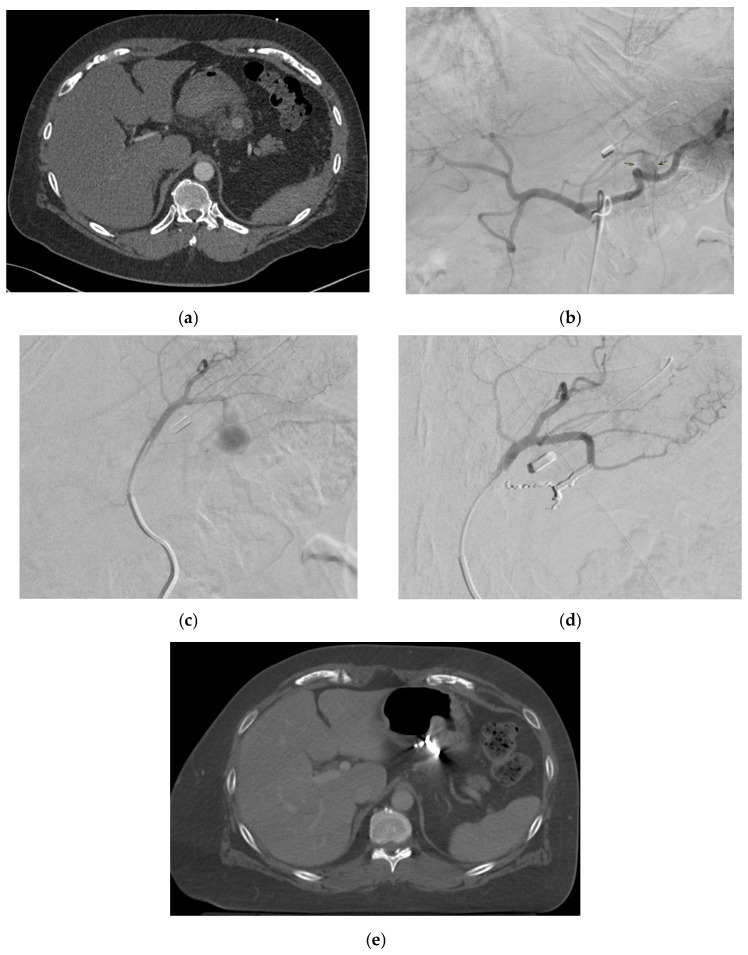
A 47-year-old male presented with abdominal pain after a motorcycle road traffic accident sustained a grade 2 splenic laceration: (**a**) CT showed a pseudoaneurysm in the region of the lesser sac with surrounding stranding. (**b**) Celiac angiogram confirmed a pseudoaneurysm arising from the left gastric artery. (**c**) This was selectively cannulated with a microcatheter and embolized successfully using coils. (**d**) Postembolization angiogram showed occlusion of the sac with no further filling. (**e**) CT follow-up done at 6 months shows the coil mass in the region of the prior pseudoaneurysm, which has since regressed.

**Figure 4 medicina-61-01312-f004:**
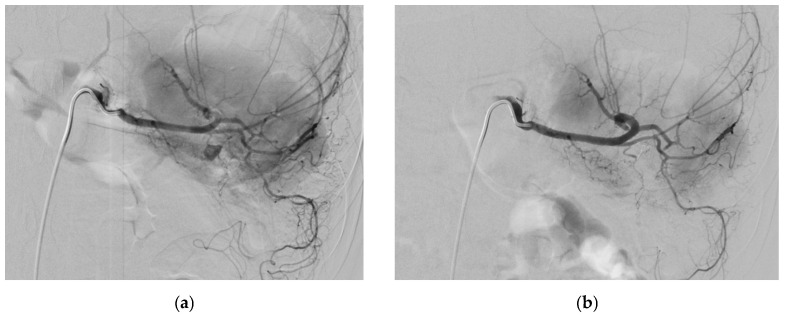
A 31-year-old man sustained grade 4 splenic laceration following a road accident: (**a**) Splenic artery angiogram shows a small pseudoaneurysm arising from an interpolar branch. This was successfully cannulated with a microcatheter and wire and eventually embolized with coils. (**b**) Postembolization angiogram shows exclusion of the pseudoaneurysm sac with preserved flow to the rest of the spleen.

**Figure 5 medicina-61-01312-f005:**
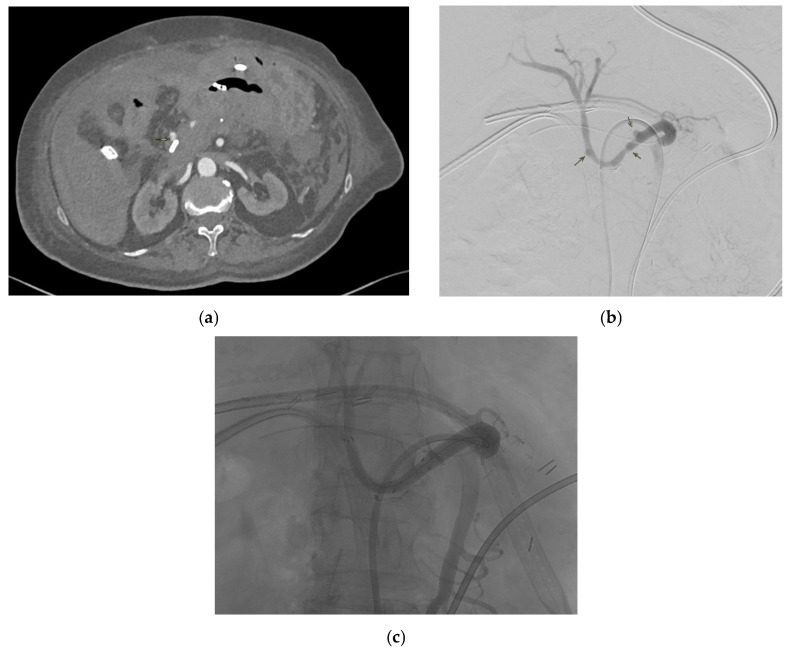
An 82-year-old female’s status following Whipple’s operation for pancreatic head malignant. Postoperative day 14, patient developed fresh bleeding from the drain with upper abdominal collection and possible postoperative pancreatic fistula: (**a**) CT angiogram shows a beaded appearance of the common hepatic artery suggestive of pseudoaneurysm formation with hemoperitoneum. (**b**) Angiogram through a sheath shows multiple pseudoaneurysms (shown by arrows) arising from the common hepatic artery. (**c**) After obtaining a wire through to the left hepatic artery, this segment was stent-grafted with Viabahn stents and molded accordingly.

**Figure 6 medicina-61-01312-f006:**
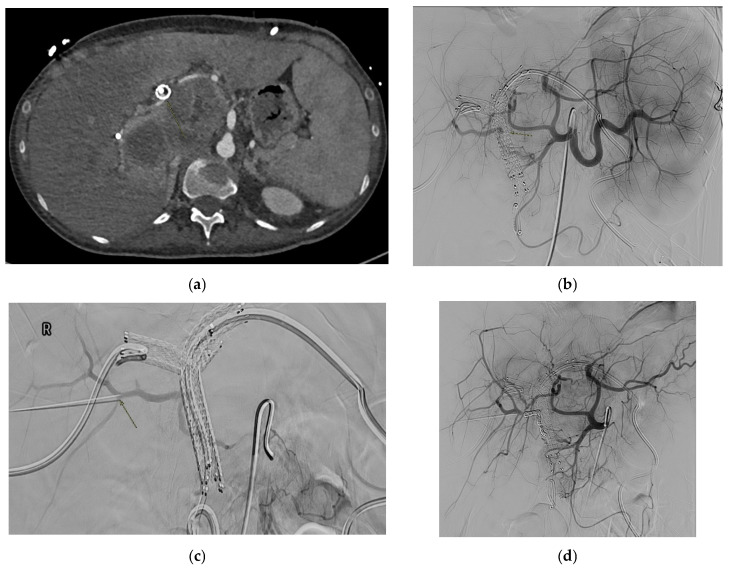
A 34-year-old lady with unresectable cholangiocarcinoma had bilateral biliary stents and drains inserted for biliary decompression for systemic chemotherapy. Patient presented with large amount of melaena: (**a**) CT angiogram shows pseudoaneurysm arising from the right hepatic artery eroding into the biliary stent. (**b**) Celiac angiogram showed the pseudoaneurysm protruding into the stent with inability to navigate through to the right hepatic artery and using a microcatheter; coils were used to embolize the pseudoaneurysm. (**c**) Ultrasound was used to target the distal right hepatic artery percutaneously. (**d**) Postembolization angiogram showed occlusion of the pseudoaneurysm, and the patient’s upper GI bleeding eventually stopped.

**Figure 7 medicina-61-01312-f007:**
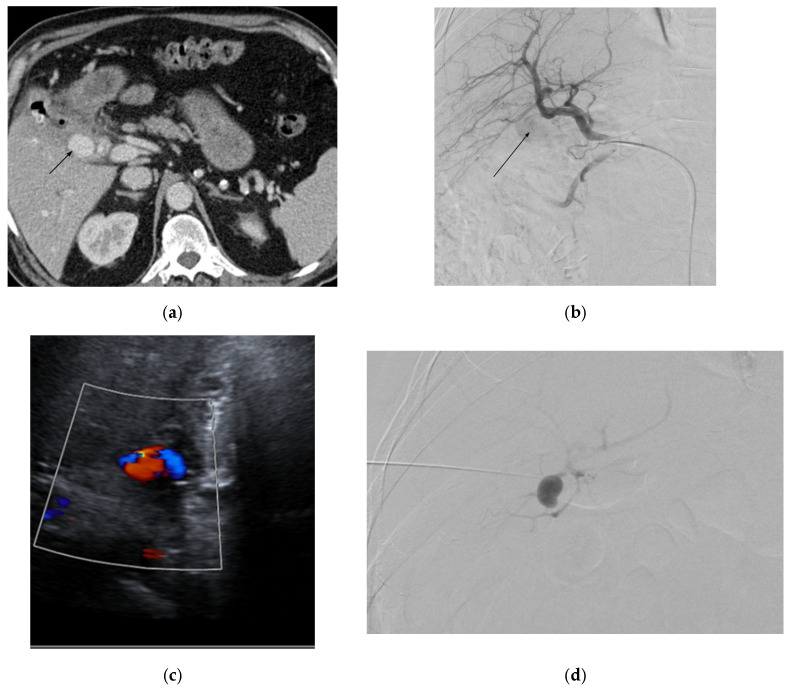
A 76-year-old gentleman with a history of chronic calculous cholecystitis presented with a history of per rectal bleeding and epigastric discomfort: (**a**) A CT scan showed possible cystic artery pseudoaneurysm (black arrow) along with chronic cholecystitis and cholecystocolonic fistula. An urgent gastroscopy was normal. A colonoscopy was performed but failed to locate the site of bleed due to poor visualization from the presence of blood clots. (**b**) Angiogram confirmed a cystic artery pseudoaneurysm (black arrow). Due to significant comorbidity and high surgical risk, a decision was made for (**c**) percutaneous Doppler-guided (**d**) thrombin embolization of the aneurysm. This was successfully performed, and no further episodes of bleeding occurred. An interval open cholecystectomy with primary colonic repair was performed.

**Figure 8 medicina-61-01312-f008:**
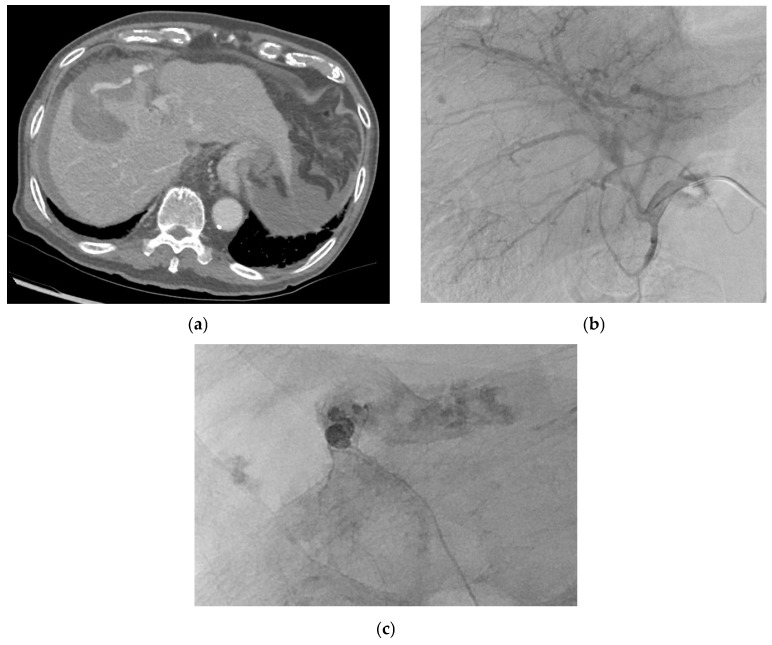
An 89-year-old gentleman with background history of atrial fibrillation on warfarin. Patient presented with severe abdominal pain: (**a**) CT on admission showed distended gallbladder with active bleeding arising from the vicinity of the gallbladder wall with hemoperitoneum. (**b**) Celiac angiogram confirmed the presence of a small pseudoaneurysm, which was eventually shown to be arising from the cystic artery. (**c**) This was successfully embolized with glue and hemostasis achieved. Patient eventually underwent a percutaneous cholecystostomy.

**Figure 9 medicina-61-01312-f009:**
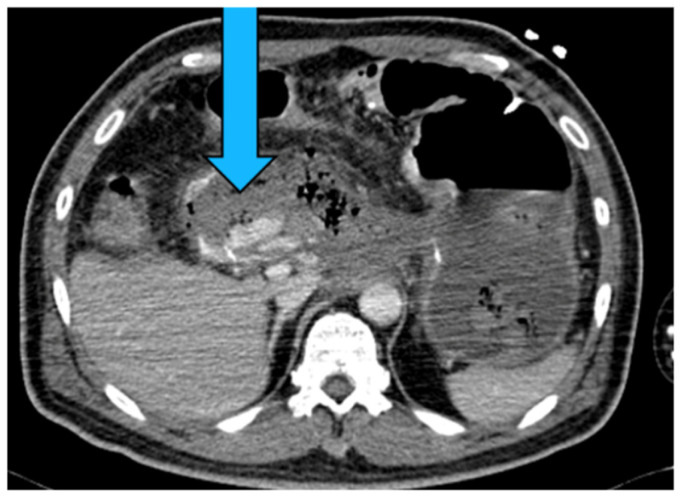
CT scan showing contrast extravasation (blue arrow) due to an iatrogenic postoperative pancreatic fistula following laparoscopic gastrectomy.

**Figure 10 medicina-61-01312-f010:**
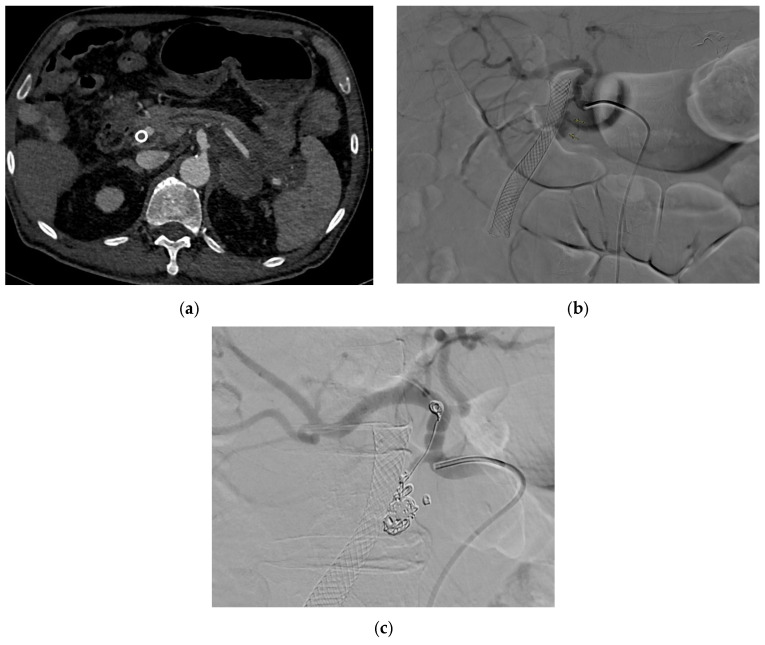
A 79-year-old male with background of metastatic pancreatic malignancy and biliary stenting presented with sepsis and gastrointestinal bleeding: (**a**) CT angiogram showed pseudoaneurysm arising adjacent to the biliary stent. (**b**) Celiac angiogram confirms a large pseudoaneurysm arising from the gastroduodenal artery. (**c**) Microcatheter was used to cannulate the sac, which was successfully embolized with coils till stasis.

**Figure 11 medicina-61-01312-f011:**
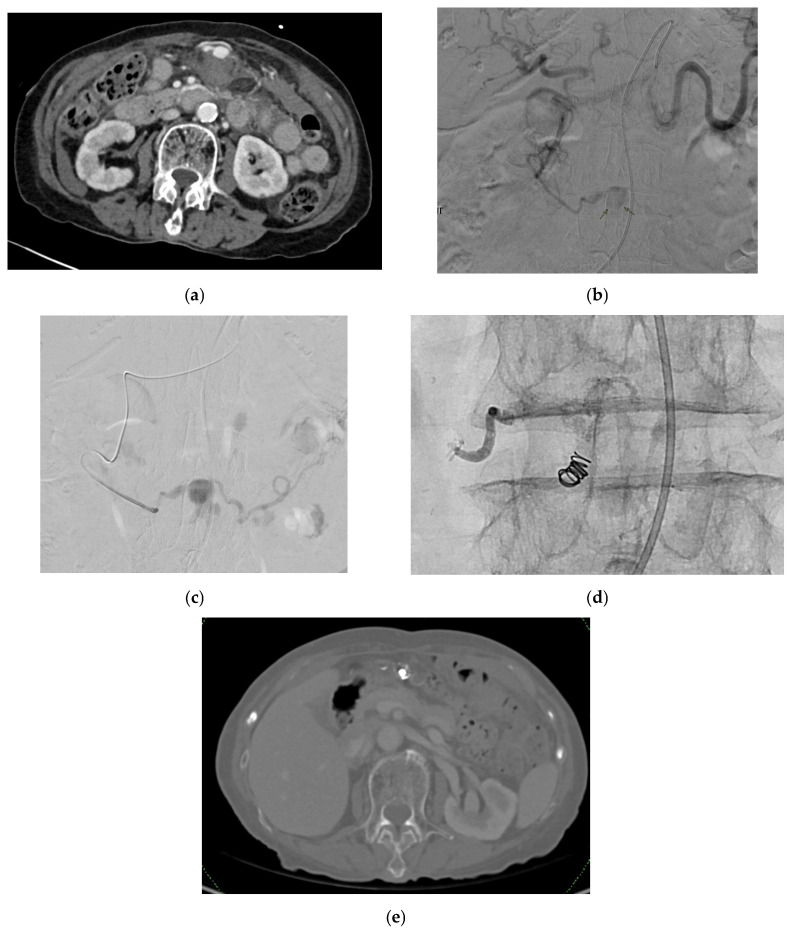
An 85-year-old female presented with sudden onset abdominal pain with vomiting, hypotension, and lactatemia: (**a**) CT shows a pseudoaneurysm arising from the right gastroepiploic artery with hemoperitoneum. (**b**) Celiac angiogram shows the pseudoaneurysm sac faintly, arising from the mid-section of the right gastroepiploic artery. (**c**) Microcatheter cannulation shows this in more detail. (**d**) The right gastroepiploic artery distal and proximal to the sac was embolized using a combination of coils and glue. (**e**) CT follow-up 9 months later for an unrelated issue shows regression of the pseudoaneurysm sac with the coil still seen within.

**Figure 12 medicina-61-01312-f012:**
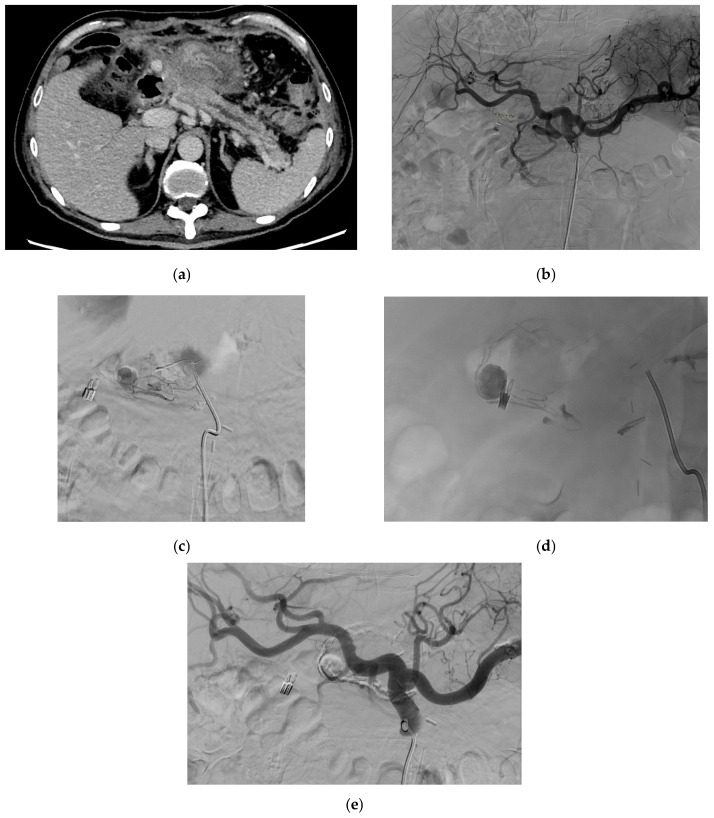
A 68-year-old male patient had a pancreatic lipomatous tumor resection. This was complicated by postoperative pancreatic fistula and a disconnected pancreatic duct, for which an endoscopic pancreatic duct stent was placed: (**a**) Patient required relaparotomies for abdominal collections and developed a pseudoaneurysm of superior pancreaticoduodenal artery which was incidentally noted on a CT scan. (**b**) Celiac angiogram confirms a rounded structure arising from superior pancreaticoduodenal artery (**c**) with Histoacryl glue embolization. (**d**) Postembolization shows glue cast within the pseudoaneurysm sac and feeding arteries. (**e**) Final angiogram shows exclusion of the pseudoaneurysm with preserved flow in the branches of the gastroduodenal and hepatic arteries.

**Figure 13 medicina-61-01312-f013:**
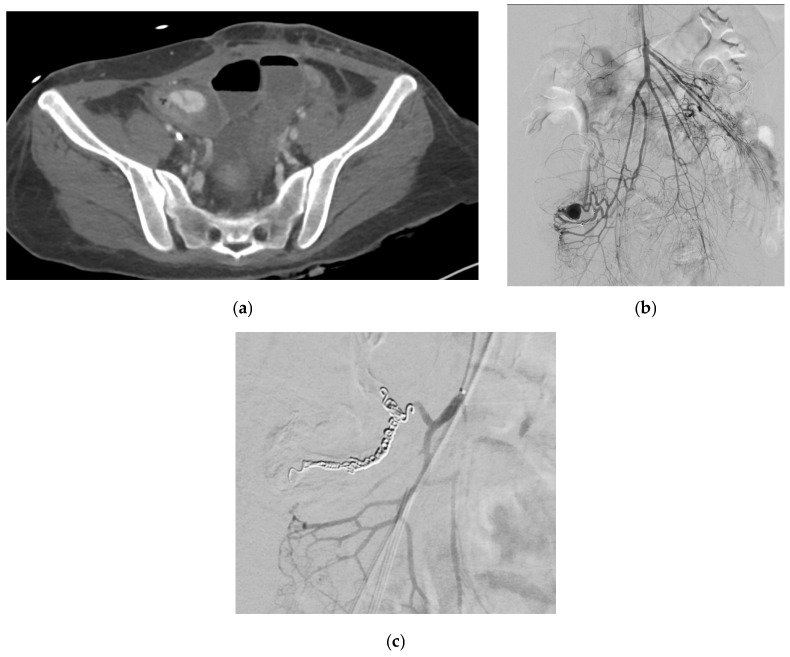
A 34-year-old lady with primary immunodeficiency disorder and multiple laparotomies, small bowel resection, and stoma formation. On the latest relook laparotomy, a short period after the operation, the patient developed large amount of fresh bleeding from the stoma: (**a**) CT angiogram confirms the presence of a pseudoaneurysm with pooling on the delayed phases and evidence of active bleeding. (**b**) SMA angiogram confirms a large pseudoaneurysm arising from an ileal artery, which was (**c**) successfully embolized with coils.

**Figure 14 medicina-61-01312-f014:**
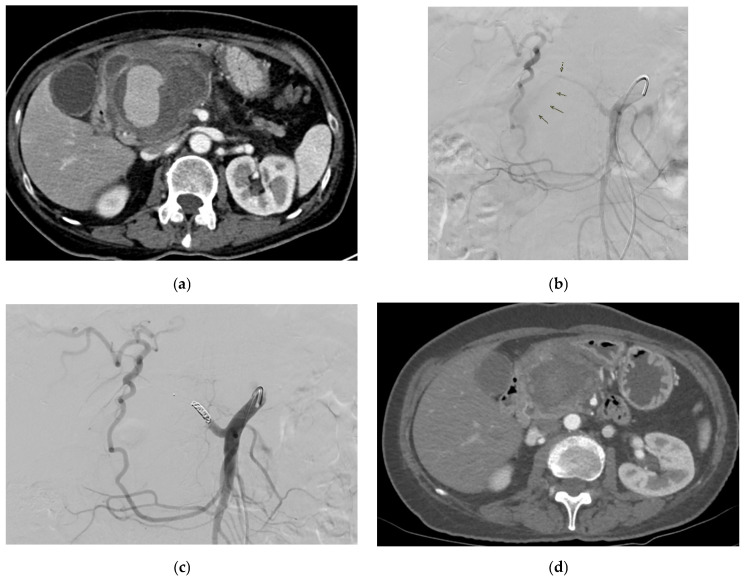
A 65-year-old female with metastatic pancreatic neuroendocrine tumor and an intratumoral pseudoaneurysm detected incidentally on CT scan: (**a**) A large intralesional pseudoaneurysm within the head of pancreas tumor. (**b**) SMA angiogram shows that the pseudoaneurysm was arising from the middle colic artery (shown by arrows), which was (**c**) successfully cannulated with a microcatheter and embolized with coils till stasis. (**d**) A CT scan follow-up 2 weeks after in the arterial phase shows no reperfusion of the pseudoaneurysm sac.

**Table 1 medicina-61-01312-t001:** Endovascular strategies for visceral artery pseudoaneurysms (VAPAs).

Aneurysm Morphology	Common Locations	Embolization Options	Technical Notes
Saccular	Splenic, GDA, SMA	Coil, vascular plug	Suitable for super-selective embolization; easy to isolate neck
Fusiform	Hepatic, Celiac	Stent graft, parent artery sacrifice	Preservation of flow preferred; stent placement may be challenging in tortuous vessels
Wide-necked	Pancreaticoduodenal, Hepatic	Balloon-assisted coiling, stent + coil	Risk of coil migration; adjunctive techniques required
Mycotic or infected	Mesenteric, colic branches	Selective embolization, followed by surgery	Definitive treatment often surgical; embolization may temporize bleeding

GDA—gastroduodenal artery, SMA—superior mesenteric artery.

## Data Availability

No new data were created or analyzed in this study. Data sharing is not applicable to this article.

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
