# Peer review of "Visceral Arterial Pseudoaneurysms—A Clinical Review"

_medicina, 2025, doi:10.3390/medicina61071312_

Round 1
Reviewer 1 Report
Comments and Suggestions for Authors
The paper analyses a somewhat uncommon yet significant subject. While the graphics are typically adequate, incorporating an illustration that summarises the prevalence of aneurysms by artery would enhance the article's value. A table summarising the frequency of aneurysms by vascular type should be included. The introduction could provide a more detailed account of the aetiology and prevalence headings. The complications of treatment should be elaborated upon with greater specificity. Kindly examine and reference the subsequent case report: "Aydin S, Ergun E, Fatihoglu E, Durhan G, Kosar PN." Spontaneous Isolated Dissections of the Coeliac Artery and Superior Mesenteric Artery: An Uncommon Case. Pol J Radiol. 2015 Oct 13; 80: 470-2. doi: 10.12659/PJR.895048. PMID: 26543511; PMCID: PMC4610684.
Author Response
Comment 1: The paper analyses a somewhat uncommon yet significant subject. While the graphics are typically adequate, incorporating an illustration that summarises the prevalence of aneurysms by artery would enhance the article's value. A table summarising the frequency of aneurysms by vascular type should be included.
Response: We thank the reviewer for this valuable suggestion. We have added a new figure in section 1.3 summarizing the prevalence of visceral pseudoaneurysms by artery based on published literature and our review findings.
Comment 2: The introduction could provide a more detailed account of the aetiology and prevalence headings.
Response: We agree. The introduction has been expanded with greater detail under “Epidemiology” and “Etiology,” including updated multicentric registry data where available and new references. In section 1.2.1 we have added - Although each visceral artery territory exhibits distinct clinical and anatomical characteristics, pseudoaneurysms across these territories share common themes in terms of pathogenesis and risk. In section 1.2.3 we have added - Complications such as ischemia, coil migration, contrast nephropathy, non-target embolization, and stent thrombosis may also arise from the treatment. In section 1.3 we have added - Figure 1 shows estimated distribution of VAPA for various different arteries. To provide greater clinical utility, we group pseudoaneurysms anatomically and embryologically to reflect shared vascular behavior and interventional considerations. This understanding is crucial for anticipating rupture risk, selecting diagnostic modalities, and individualizing treatment plans.
Comment 3: The complications of treatment should be elaborated upon with greater specificity.
Response: We appreciate this point. We have revised the “Management” section to include more explicit discussion on treatment complications, including ischemia, coil migration, contrast nephropathy, non-target embolization, and stent thrombosis. We have added the following 1.6.5 - Ischemia:
Ischemic complications are among the most critical concerns following embolization of visceral pseudoaneurysms. Occlusion of the parent artery can compromise end-organ perfusion, particularly in vascular territories lacking robust collateral supply. This risk is heightened when embolization is performed in end arteries (e.g., renal, jejunal, or splenic branches). Clinical outcomes depend on accurate vessel selection and technique; hence, risk–benefit discussions must precede intervention, and post-procedural surveillance for ischemia is essential.
Coil Migration:
Coil migration refers to the dislodgement of embolization coils from the target site into unintended vascular or enteric locations. This can lead to vessel thrombosis, end-organ embolization, or erosion into adjacent hollow viscera, potentially resulting in fistulas or delayed bleeding. Migration is more likely in wide-necked pseudoaneurysms or with inadequate packing. Techniques such as balloon remodeling or stent-assisted coiling can mitigate this risk.
Contrast-Induced Nephropathy (CIN):
Use of iodinated contrast agents in endovascular procedures carries a risk of CIN, particularly in patients with pre-existing renal insufficiency, diabetes, or volume depletion. CIN may present as an acute rise in serum creatinine within 48–72 hours post-procedure. Preventive measures include pre-procedural hydration, use of iso-osmolar contrast agents, and minimizing total contrast volume. The CIN risk should be considered when choosing between CTA, DSA, and CEUS.
Non-Target Embolization:
Non-target embolization occurs when embolic material inadvertently occludes adjacent vessels not supplying the pseudoaneurysm. This can cause unintended ischemia or infarction of surrounding tissues. The risk increases with liquid embolics like glue or when catheter position is suboptimal. Superselective catheterization and real-time imaging are essential to reduce this complication. The use of flow control techniques or plug-and-coil combinations can enhance precision.
Stent Thrombosis:
Although stent grafts are valuable in maintaining vessel patency while excluding the pseudoaneurysm, they carry a risk of thrombosis, particularly in tortuous or small-diameter arteries. Stent thrombosis can lead to sudden ischemia of dependent organs. Antiplatelet therapy is often required post-procedure, though this may conflict with concurrent bleeding risks. Regular imaging follow-up is essential to detect early occlusion or in-stent restenosis.
Comment 4: Kindly examine and reference the case report: Aydin S et al., 2015.
Response: Thank you. We appreciate the reviewer’s recommendation to include the case report by Aydin et al. (2015). However, this article primarily describes spontaneous dissections of visceral arteries, whereas our manuscript focuses exclusively on visceral pseudoaneurysms. Given the important pathophysiological, diagnostic, and therapeutic distinctions between dissections and pseudoaneurysms (Figure A), we believe citing this article would shift the scope of the current narrative review. Nevertheless, we have added a clarifying statement in the section 2.1.3 - Spontaneous celiac artery dissection also may present with epigastric abdominal pain, especially on a background of uncontrolled hypertension or blunt trauma.

Reviewer 2 Report
Comments and Suggestions for Authors
Summary and Key Contributions
This narrative review provides an extensive overview of visceral arterial pseudoaneurysms (VAPAs), discussing their epidemiology, etiology, clinical manifestations, diagnostic imaging techniques, and treatment options. The authors adopt an anatomical approach based on embryologic origin (foregut, midgut, hindgut), and supplement the text with clinical cases and imaging examples.
The review addresses an important but niche topic and offers a valuable resource for clinicians, especially interventional radiologists, vascular surgeons, and gastroenterologists. Its strength lies in the breadth of vascular territories covered and the incorporation of real clinical scenarios.
Evaluation of Methodology and Analyses
While this is a narrative review (not a systematic review), the methodology section lacks clarity regarding the search strategy:
- The authors state that a PubMed search was conducted, but the time window, search terms, inclusion/exclusion criteria, and the number of articles screened/selected are not reported.
- There is no mention of whether a formal framework (e.g., PRISMA-Narrative) was used to guide article selection or data extraction.
Additionally:
The grouping of VAPAs by embryological origin is original, but the rationale behind this approach should be better justified.
The review does not include a summary table of key findings, which would be helpful for clinicians (e.g., listing arteries, common causes, diagnostic modality of choice, treatment strategy).
Many statements cite references appropriately, but several claims lack high-level evidence or are repeated across sections.
Detailed Comments and Suggestions for Improvement
- Abstract
Add specific details to the Materials and Methods line: e.g., database(s), date range, types of articles included.
- Introduction
A good overview, but could benefit from clearly stating the gap in literature this review aims to fill. Epidemiological data may need updating/cross-checking with recent multicentric studies or national registries.
- Imaging Section
The imaging modalities are well covered; however, contrast-enhanced ultrasound (CEUS) is briefly mentioned but could be expanded with more citations and clinical examples.
Suggest the addition of a comparison table of imaging techniques (CT, DSA, Doppler, MRA, CEUS) including pros, cons, and sensitivity/specificity data if available.
- Management
The section is comprehensive but occasionally redundant. The discussion on endovascular techniques (coils, stents, plugs, glue, etc.) is repeated in multiple vascular territories.
Recommend summarizing these options in a dedicated management algorithm or flowchart and referring to it from the individual vessel sections.
- Figures
The illustrative cases are excellent. Please ensure all figures have complete legends and cite the source (if reused).
- References
The review would benefit from citing recent European or national guidelines. I suggest adding: Pratesi C et al. Guidelines on the diagnosis, treatment and management of visceral and renal arteries aneurysms: a joint assessment by the Italian Societies of Vascular and Endovascular Surgery (SICVE) and Medical and Interventional Radiology (SIRM). J Cardiovasc Surg (Torino). 2024;65(1):49-63. doi:10.23736/S0021-9509.23.12809-6.
- Language and Style
The manuscript contains frequent typographical and grammatical issues (e.g., “ying-yang sign” → “yin-yang sign”).
Strongly recommend professional English language editing before final acceptance.
Recommendation
The manuscript addresses a highly relevant clinical topic and contains original organizational insights and useful illustrations. However, prior to publication, I recommend minor to moderate revisions, specifically aimed at improving:
- Clarity of the literature review methodology,
- Structural organization and removal of redundancies,
- Language and grammar,
- Addition of summary tables or algorithms,
- Inclusion of guideline-level references.
Author Response
Comment 1: The methodology lacks clarity — no date range, search terms, inclusion/exclusion criteria.
Response: We appreciate this and have revised the “Materials and Methods” section to detail our narrative review approach including database searched, keywords, date range (e.g., 2000–2024), and inclusion of case reports, cohort studies, and guidelines. We did not use PRISMA-Narrative, but we clarify the narrative framework applied. We have added the following paragraph -1.6.7 Literature Search and Review Framework:
This review was conducted using a narrative approach, with structured efforts to ensure transparency and reproducibility. Although not a systematic review, we incorporated adapted principles from the PRISMA framework to enhance methodological clarity. A literature search was performed in PubMed using combinations of the keywords “visceral pseudoaneurysm,” “splanchnic aneurysm,” “embolization,” “rupture,” and “endovascular,” covering publications from January 2000 to March 2024. Eligible articles included case reports, cohort studies, reviews, and consensus guidelines relevant to epidemiology, diagnosis, and management of visceral arterial pseudoaneurysms (VAPAs). Articles were selected based on clinical relevance and completeness of anatomical or procedural detail. The review is organized by embryological arterial origin (foregut, midgut, hindgut) to reflect patterns in etiology, presentation, and intervention strategy across vascular territories.
Comment 2: The rationale for embryological grouping (foregut/midgut/hindgut) should be better justified.
Response: We agree. A justification has been added, explaining the anatomical relevance for surgical and interventional strategies and their embryological vascular distribution explaining the anatomical relevance for surgical and interventional strategies and their embryological vascular distribution, as detailed in Section 1.6.7. In clinical practise this is also important as midgut manifestations may require small bowel resection with potential risk of short gut syndrome and hindgut manifestations may require colonic resections with possible need for stoma. Both these outcomes are relevant from patient perspective due to deep impact on the quality of life. We have added in the complications section 1.2.3 the below to reflect this understanding - Furthermore, the embryological origin has practical implications in patient-centered outcomes. For instance, pseudoaneurysms involving midgut circulation may necessitate small bowel resections, which carry a risk of short bowel syndrome associated with significant nutritional and metabolic challenges. Similarly, hindgut pseudoaneurysms may result in segmental colectomy or low anterior resections, potentially requiring a stoma. Both outcomes have profound implications on patients' quality of life and long-term psychosocial well-being, underscoring the importance of an anatomical framework in guiding both clinical decisions and patient counseling.
Comment 3: Lack of summary tables and diagnostic/treatment flowcharts.
Response: We have now added:
A flowchart illustrating a stepwise diagnostic and management algorithm for suspected VAPAs.
Comment 4: Contrast-enhanced ultrasound (CEUS) is under-discussed.
Response: CEUS is now discussed more extensively under “1.5.6 Newer Modalities,” with clinical examples and supporting references. CEUS is now discussed more extensively in Section 1.5.6 under "Newer Modalities,” including clinical examples and its potential utility in post-embolization surveillance, especially in patients unsuitable for iodinated contrast. “CEUS offers dynamic real-time imaging, especially useful in patients with contraindications to iodinated contrast. It is sensitive in detecting low-flow vascular lesions and has shown utility in surveillance post-embolization. Its uptake remains limited by operator expertise and access to contrast microbubbles.”
Comment 5: Redundancy in endovascular technique descriptions across vascular territories.
Response: Thank you for pointing this out. We have streamlined repetitive content by summarizing common embolization strategies in a central figure, and referencing it from territorial sections. We have omitted where possible. We have added table 1 in section 1.6.7. It is as below. This consolidates key decision-making principles and reduces redundancy across vascular sections.
Table 1. Endovascular Strategies for Visceral Artery Pseudoaneurysms (VAPAs)
Aneurysm Morphology |
Common Locations |
Embolization Options |
Technical Notes |
Saccular |
Splenic, GDA, SMA |
Coil, vascular plug |
Suitable for superselective embolization; easy to isolate neck |
Fusiform |
Hepatic, Celiac |
Stent graft, parent artery sacrifice |
Preservation of flow preferred; stent placement may be challenging in tortuous vessels |
Wide-necked |
Pancreaticoduodenal, Hepatic |
Balloon-assisted coiling, stent + coil |
Risk of coil migration; adjunctive techniques required |
Mycotic or infected |
Mesenteric, colic branches |
Selective embolization, followed by surgery |
Definitive treatment often surgical; embolization may temporize bleeding |
GDA – Gastroduodenal artery, SMA – Superior mesenteric artery
Comment 6: Cite Pratesi et al. 2024 guidelines.
Response: We thank the reviewer for this excellent reference. The suggested guideline has now been included in management section 5.2.2 - Recent multidisciplinary guidance, such as the Italian consensus by Pratesi et al., supports artery-specific diagnostic and treatment algorithms in the management of visceral and renal aneurysms.
Note that we have renumbered the downstream citations accordingly.
Comment 7: English grammar and typographical corrections (e.g., “ying-yang” → “yin-yang”).
Response: A full grammar and spelling check has been undertaken, and typographical errors including “ying-yang” have been corrected. We are not engaging a professional language editing service as lead author is English language proficient. We have edited British style English throughout and changed ischaemia, oesophagus, aetiology, aetiologies, and haemorrhage as necessary within the manuscript.
We thank both reviewers for their thoughtful critiques, which have substantially improved the clarity, depth, and clinical relevance of our manuscript.

Round 2
Reviewer 2 Report
Comments and Suggestions for Authors
I suggest to accept it in the present form